# Conformal prediction for causal effects of continuous treatments

## Abstract

Uncertainty quantification of causal effects is crucial for safety-critical applications such as personalized medicine. A powerful approach for this is conformal prediction, which has several practical benefits due to model-agnostic finite-sample guarantees. Yet, existing methods for conformal prediction of causal effects are limited to binary/discrete treatments and make highly restrictive assumptions such as known propensity scores. In this work, we provide a novel conformal prediction method for potential outcomes of continuous treatments. We account for the additional uncertainty introduced through propensity estimation so that our conformal prediction intervals are valid even if the propensity score is unknown. Our contributions are three-fold: (1) We derive finite-sample validity guarantees for prediction intervals of potential outcomes of continuous treatments. (2) We provide an algorithm for calculating the derived intervals. (3) We demonstrate the effectiveness of the conformal prediction intervals in experiments on synthetic and real-world datasets. To the best of our knowledge, we are the first to propose conformal prediction for continuous treatments when the propensity score is unknown and must be estimated from data.

## 1 Introduction

Machine learning (ML) for estimating causal quantities such as causal effects and the potential outcomes of treatments is nowadays widely used in real-world applications such as personalized medicine (Feuerriegel et al., 2024). However, existing methods from causal ML typically focus on point estimates (e.g., Nie et al., 2021; Patrick Schwab et al., 2020), which means that the uncertainty in the predictions is neglected and hinders the use of causal ML in safety-critical applications (Feuerriegel et al., 2024; Kneib et al., 2023). As the following example shows, uncertainty quantification (UQ) of causal quantities is crucial for reliable decision-making.

*Motivating example:* Let us consider a doctor who seeks to determine the dosage of chemotherapy in cancer care. This requires estimating the tumor size in response to the dosage for a specific patient profile. A point estimate will predict the *average* size of the tumor post-treatment, but it will neglect that chemotherapy is ineffective for some patients. In contrast, UQ will give a *range* of the tumor size that is to be expected post-treatment, so that doctors can assess the probability that the patients will actually benefit from treatment. This helps to understand the risk of a treatment being ineffective and can guide doctors to choose treatments that are effective *with large probability*.

A powerful method for UQ is ***conformal prediction*** (CP) (Lei & Wasserman, 2014; Papadopoulos, 2002; Vovk et al., 2005). CP provides model-agnostic and distribution-free, finite-sample validity guarantees for quantifying uncertainty. CP has been widely used for traditional, predictive ML (e.g., Angelopoulos et al., 2024; Barber et al., 2023; Gibbs et al., 2023), where it has been shown to yield reliable prediction intervals in finite-sample settings (see Fig. 1). Recently, there have been works that adapt CP for estimating causal quantities (see

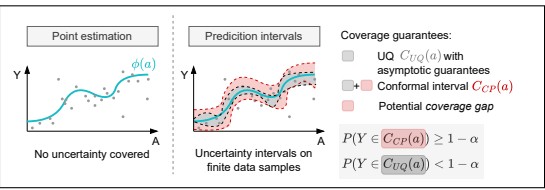

Figure 1: CP intervals on finite-sample data. UQ methods with asymptotic guarantees might suffer from under-coverage and are often *not* faithful. Thus, we aim at CP with finite-sample guarantees.

Fig. 2 for an overview). Yet, existing methods for CP focus on binary or discrete treatments (e.g., Alaa et al., 2023; Jonkers et al., 2024; Lei & Candès, 2021), but *not* continuous treatments-

Adapting CP to causal quantities is *non-trivial* for two main reasons. **Challenge (a)**: Intervening on the treatment induces a shift in the covariate distribution, specifically, in the propensity score. As a result, the so-called *exchangeability assumption*, which is inherent to CP (Vovk et al., 2005), is violated between the observational and interventional distribution, and because of this, standard CP intervals are not valid. Thus, we must later account for the distribution shift and derive *treatment-conditional guarantees*. **Challenge (b)**: Assessing the aforementioned shift in the distribution requires information about the propensity score; yet, the propensity score is typically unknown. Hence, estimating the propensity score introduces *additional uncertainty*. However, incorporating the additional uncertainty in the overall CP intervals cannot be done in a simple plug-in manner, and it is highly non-trivial.

Unique to CP for effects of continuous treatments is a third challenge (a): data points with the same treatment value are rarely observed. Thus, we later employ smoothing to model the propensity shift.

In this paper, we develop a CP method for causal quantities, such as potential outcomes of continuous treatments. Our method is designed to account for the additional uncertainty introduced during propensity estimation and is thus applicable to settings where the propensity score is known or unknown.

**Our contributions:**[1] (1) We propose a novel method for CP of causal quantities such as potential outcomes or treatment effects of continuous treatments. For this, we mathematically derive finite-sample prediction intervals for potential outcomes under known and unknown propensity functions. (2) We provide an algorithm for efficiently calculating the derived intervals. (3) We demonstrate the effectiveness of the derived CP intervals in experiments on multiple datasets.

## 2 RELATED WORK

**UQ for causal effects:** Existing methods for UQ of causal quantities are often based on Bayesian methods (e.g., Alaa & van der Schaar, 2017; Hess et al., 2024; Hill, 2011; Jesson et al., 2020). However, Bayesian methods require the specification of a prior distribution based on domain knowledge and are thus neither robust to model misspecification nor generalizable to model-agnostic machine learning models. A common ad hoc method for computing uncertainty intervals is Monte Carlo (MC) dropout (Gal & Ghahramani, 2016). However, MC dropout yields approximations of the posterior distribution, which are *not* faithful (Le Folgoc et al., 2021).

**Conformal prediction:** CP (Lei & Wasserman, 2014; Papadopoulos, 2002; Vovk et al., 2005) has recently received large attention for finite-sample UQ. For a prediction model $\phi$ trained on dataset $D_T = (X_i, Y_i)_{i=1,\dots,m}$ and a new test sample $X_k$, CP aims to construct a prediction interval $C(X_k)$ such that $P(Y_k \in C(X_k)) \geq 1 - \alpha$ for some significance level $\alpha$. We refer to (Angelopoulos et al., 2024) for an in-depth overview. Due to its strong finite-sample validity guarantees, CP is widely used for traditional, predictive ML with widespread applications such as in medical settings (Zhan et al., 2020) or drug discovery (Alvarsson et al., 2021; Eklund et al., 2015).

Several extensions have been developed for CP. One literature stream focuses on CP with *marginal coverage* under distribution shifts between training and test data (e.g., Cauchois et al., 2020; Fannjiang et al., 2022; Gendler et al., 2022; Ghosh et al., 2023; Gibbs & Candès, 2021; Gibbs et al., 2023; Guan, 2023; Lei & Candès, 2021; Podkopaev & Ramdas, 2021; Tibshirani et al., 2019; Yang et al., 2024). Our setting later also involves a distribution shift due to the intervention on the treatment but differs from the latter in that the true distribution shift is unknown. Another literature stream constructs intervals *conditional* on the variables following the shifted distribution. Since, in general, exact conditional coverage has been proven impossible (Lei & Wasserman, 2014; Vovk, 2012), the works in this literature stream have two key limitations: (1) they only guarantee *approximate* conditional coverage (e.g., Barber et al., 2021; Cai et al., 2014; Lei & Wasserman, 2014; Romano et al., 2020); or (2) they are restricted to specific data structures such as binary variables (e.g., Lei & Wasserman, 2014; Vovk, 2012). Because of that, none of the existing methods for marginal and conditional coverage can be applied to derive prediction intervals with finite-sample validity guarantees for causal quantities of continuous treatments.

---

[1]Code and data are available at our public GitHub repository: `https://anonymous.4open.science/r/CausalConformalPrediction_anonymous-B730/README.md`

**Conformal prediction for causal quantities:** Only a few works focus on CP for causal quantities (see Fig. 2). Examples are methods aimed at off-policy learning (Taufiq et al., 2022; Zhang et al., 2023), conformal sensitivity analysis (Yin et al., 2022), or meta-learners for the conditional average treatment effect (CATE) (Alaa et al., 2023; Jonkers et al., 2024; Lei & Candès, 2021; Wang et al., 2024). However, there are crucial differences to our setting: First, the existing works (a) assume that the propensity is *known* and thus achieve finite-sample coverage guarantees, or the existing works (b) focus on the easier task of giving *asymptotic* guarantees but then might suffer from under-coverage because of which the intervals are *not* faithful. Only Lei & Candès (2021) provides finite-sample coverage guarantees under estimated propensity scores. However, all existing CP methods are designed for *binary* or *discrete* treatments. Applying such methods to discretized continuous treatments leads to ill-defined causal estimands. Therefore, none of the existing methods are applicable to our continuous treatment setting. We offer a detailed discussion in Supplement E.

**Research gap:** To the best of our knowledge, no work has provided prediction intervals with finite-sample validity guarantees for causal quantities of continuous treatments.

| Causal conformal prediction | Unknown propensity | Finite sample exact guarantees | Continuous treatment |
|---|---|---|---|
| e.g., Alaa et al. (2023), Wang et al. (2024) | ✗ | ✓ | ✗ |
| e.g., Jin et al. (2023), Jonkers et al. (2024) | ✓ | ✗ | ✗ |
| Lei and Candès (2021) | ✓ | ✓ | ✗ |
| **Ours** | ✓ | ✓ | ✓ |

Figure 2: Key works on CP in causal inference.

## 3 PROBLEM FORMULATION

**Notation:** We denote random variables by capital letters $X$ with realizations $x$. Let $P_X$ be the probability distribution over $X$. We omit the subscript whenever it is obvious from the context. For discrete $X$, we denote the probability mass function by $P(x) = P(X = x)$ and the conditional probability mass functions by $P(y \mid x) = P(Y = y \mid X = x)$ for a discrete random variable $Y$. For continuous $X$, $p(x)$ is the probability density function w.r.t. the Lebesgue measure.

**Setting:** Let the data $(X_i, A_i, Y_i)_{i=1,\dots,n}$ consisting of observed confounders $X \in \mathcal{X}$, a continuous treatment $A \in \mathcal{A}$, and an outcome $Y \in \mathcal{Y}$ be drawn *exchangeably* from the joint distribution $P$. Additionally, let a new sample of confounders $X_{n+1}$ be drawn independently from the marginal distribution $P_X$. Throughout our work, we split the dataset into a proper training dataset $D_T = (X_i, A_i, Y_i)_{i=1,\dots,m}$, and a calibration dataset $D_C = (X_i, A_i, Y_i)_{i=m+1,\dots,n}$. Furthermore, let $\pi(a \mid x)$ define the generalized propensity score for treatment $A = a$ given $X = x$.

Throughout this work, we build upon the potential outcomes framework (Rubin, 2005). We denote the potential outcomes after a hard intervention $a^*$ by $Y(a^*)$ and after a soft intervention $A^*(x) \sim \tilde{\pi}(a \mid x) = P_{A^* \mid X = x}$ by $Y(A^*(x))$.[2] We make three standard identifiability assumptions for causal effect estimation: positivity, consistency, and unconfoundedness (e.g., Alaa et al., 2023; Jonkers et al., 2024). Finally, we consider an arbitrary machine learning model $\phi$ to predict the potential outcomes. Hence, we define the outcome prediction function as $\phi : \mathcal{X} \times \mathcal{A} \to \mathbb{R}, \phi(X, A) \mapsto Y$. We assume the dose-response curve to be sufficiently smooth. This is common in causal inference with continuous treatments (e.g., Patrick Schwab et al., 2020; Schweisthal et al., 2023).

**Our objective:** In this work, we aim to derive *conformal prediction intervals* $C(X_{n+1}, \Diamond)$ for the prediction of a potential outcome $Y_{n+1}(\Diamond)$ of a new data point under either hard, $\Diamond = a^*$, or soft intervention, $\Diamond = A^*(X_{n+1}) \sim \tilde{\pi}(a \mid X_{n+1})$. The derived intervals are called *valid* for any new exchangeable sample $X_{n+1}$ with non-exchangeable intervention $\Diamond$, i.e.,

$$P(Y_{n+1}(\Diamond) \in C(X_{n+1}, \Diamond)) \geq 1 - \alpha, \quad \Diamond \in \{a^*, A^*(X_{n+1})\}. \tag{1}$$

for some significance level $\alpha \in (0, 1)$. Of note, our CP method can later be used with an arbitrary machine learning model $\phi$ for predicting the potential outcomes.

In CP, the interval $C$ is constructed based on so-called *non-conformity scores* (Vovk et al., 2005), which capture the performance of the prediction model $\phi$. For example, a common choice for the

---

[2]Interventions are characterized by two classes: hard (structural) and soft (parametric) interventions. Hard interventions directly affect the treatment by setting it to a specific value and removing the edge in the graph (as in the do-operator). Soft interventions do not change the structure of the graph but affect the conditional distribution of the treatment given the confounders, i.e., the propensity score.

non-conformity score is the residual of the fitted model $s(X, A, Y) = |Y - \phi(X, A)|$, which we will use throughout our work. For ease of notation, we define $S_i := s(X_i, A_i, Y_i)$.

**Why is CP for causal quantities non-trivial?** There are two main reasons. First, coverage guarantees of CP intervals essentially rely on the exchangeability of the non-conformity scores. However, intervening on treatment $A$ shifts the propensity function and, therefore, induces a shift in the covariates $(X, A)$ ($\rightarrow$ Challenge (a)). Formally, we have a *propensity shift* in which the intervention $\diamondsuit$ shifts the propensity function $\pi(a \mid x)$ to either a Dirac-delta distribution of the hard intervention, $\delta_{a^*}(a)$, or to the distribution of the soft intervention, $\tilde{\pi}(a \mid x)$, without affecting the outcome function $\phi(x, a)$. As a result, the test data sample under $\diamondsuit$ does *not follow the same distribution* as the train and calibration data, i.e., the exchangeability assumption is violated.

Second, the propensity score $\pi$ is commonly *unknown* in observational data and, therefore, must be estimated, which introduces additional uncertainty that one must account for when constructing CP intervals ($\rightarrow$ Challenge (b)). Crucially, existing coverage guarantees (e.g., Vovk, 2012; Tibshirani et al., 2019) do *not* hold in our setting. Instead, we must derive new intervals with valid *coverage under propensity shift*.

In the following section, we address the above propensity shift by performing a calibration conditional on the propensity shift induced by the intervention, which allows us then to yield valid prediction intervals with significance level $(1 - \alpha)$ for potential outcomes of a specific hard or soft intervention. We derive the potential outcomes and emphasize that the extension to causal effects is straightforward. For the latter, one combines the intervals for each potential outcome under a certain treatment and without treatment, so that eventually arrives at CP intervals for the individual treatment effect (ITE). Details are Supplement A.

## 4 CP INTERVALS FOR POTENTIAL OUTCOMES OF CONTINUOUS TREATMENTS

Recall that intervening on test data samples breaks the exchangeability assumption necessary for the validity, i.e., the guaranteed coverage of at least $(1 - \alpha)$, of CP intervals. Therefore, we now construct CP intervals where we account for a (potentially unknown) covariate shift in the test data induced by the intervention.

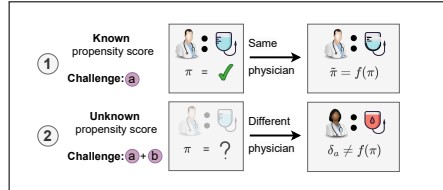

**Scenarios:** In our derivation, we distinguish two different scenarios(see Fig. 3):

Figure 3: Use cases of the two scenarios: (1) The new assignment is a function of the original policy (i.e., soft intervention). (2) The policy in the dataset is unknown. The new assignment cannot be expressed as a function of the original policy (i.e., hard intervention).

(1) **Known propensity score** (see Section 4.1): If the propensity score in the observational data is known, it means that the treatment policy is known. Then, we aim to update the policy by increasing/decreasing the treatment by a value $\Delta_A$, i.e., $A^*(X) = A + \Delta_A$.

*Example:* Imagine a doctor is about to prescribe a medication to a new patient. Instead of prescribing the same dosage as he would have prescribed to a similar patient in the past, the doctor is interested in the potential health outcome of the patient when increasing (or decreasing) the original dosage by amount $\Delta_A$.

(2) **Unknown propensity score** (see Section 4.2): In observational data, the propensity score is typically unknown. Therefore, we usually are interested in the effect of hard interventions, i.e., $a^*$. Here, we face additional uncertainty from the propensity score estimation ($\rightarrow$ Challenge (b)).[3]

*Example:* In our running example, a patient comes to a new doctor who has never prescribed the respective medication and thus will base the decision on observational data (electronic health records) from other physicians, yet the observational data was collected under a different, unknown treatment policy. Therefore, the prescribed intervention (and thus dosage) cannot be expressed in terms of the policy in the observational data.

---

[3]Throughout our main paper, we focus on the setting of hard interventions. In some cases it might also be of interest to perform soft interventions on the estimated propensity score. We provide derivations for this setting in Supplement A.

In our derivations, we make use of the following two mathematical tools. First, we define the *propensity shift*. Formally, it is the shift between the observational and interventional distributions, $P$ and $\tilde{P}$, in terms of the tilting of the propensity function by a non-negative function $f$. Hence, we have

$$\tilde{\pi}(a \mid x) = \frac{f(a, x)}{\mathbb{E}_P[f(A, X)]} \pi(a \mid x). \tag{2}$$

for some $f$ with $\mathbb{E}_P[f(X, A)] > 0$ and $a \in \mathcal{A}, x \in \mathcal{X}$.

Second, our CP method will build upon ideas from so-called split conformal prediction (Papadopoulos, 2002; Vovk et al., 2005), yet with crucial differences. In our methods, the calibration step differs from the standard procedure in that we *conditionally calibrate* the non-conformity scores depending on the tilting function $f$ to achieve marginal coverage for the interventional – and thus shifted – data.

**High-level outline:** Our derivation in Sections 4.1 and 4.2 proceed as follows. Following (Gibbs et al., 2023; Romano et al., 2019), we reformulate split conformal prediction as an augmented quantile regression. Let $S_i$ represent the non-conformity score of the sample $(X_i, A_i, Y_i)$ for $i = m+1, \ldots, n$ of the calibration dataset and $S_{n+1} = S$ an imputed value for the unknown score of the new sample. We define

$$\hat{\theta}_S := \arg\min_{\theta \in \mathbb{R}} \frac{1}{n-m} \left( \sum_{i=m+1}^{n} l_\alpha(\theta, S_i) + l_\alpha(\theta, S) \right), \tag{3}$$

where

$$l_\alpha(\theta, S) := \begin{cases} (1-\alpha)(S - \theta), & \text{if } S \geq \theta, \\ \alpha(\theta - S), & \text{if } S < \theta. \end{cases} \tag{4}$$

Of note, $\hat{\theta}_S$ is an estimator of the $(1-\alpha)$-quantile of the non-conformity scores (Koenker & Bassett, 1978; Steinwart & Christmann, 2011). Using $\hat{\theta}_S$, we then construct the CP interval with the desired coverage guarantee $(1-\alpha)$. However, the interval is only valid for exchangeable data. Quantile regression might yield non-unique solutions that can depend on the indices of the scores (Gibbs et al., 2023), so we later restrict the analysis to solvers invariant to the data ordering.[4]

### 4.1 Scenario 1: Known propensity score

We first consider scenario ① with known propensity scores. Here, existing CP intervals are not directly applicable due to the shift from old to new propensity ($\rightarrow$ Challenge ⓐ). For our derivation, we need the following lemma building upon and generalizing the intuition presented above.

**Lemma 1** ((Gibbs et al., 2023)). *Let $\mathcal{F}$ define a finite-dimensional function class that includes the function $f$ characterizing the shift in the (potentially unknown) propensity function $\pi$ (see Eq. equation 2). Define the distribution-shift-calibrated $(1-\alpha)$-quantile of the non-conformity scores as*

$$\hat{g}_S(X_{n+1}) := \arg\min_{g \in \mathcal{F}} \frac{1}{n-m} \left( \sum_{i=m+1}^{n} l_\alpha(g(X_i), S_i) + l_\alpha(g(X_{n+1}), S) \right) \tag{5}$$

*for an imputed guess $S$ of the $(n+1)$-th non-conformity score $S_{n+1}$. The prediction interval*

$$C(X_{n+1}) := \{y \mid S_{n+1}(y) \leq \hat{g}_{S_{n+1}(y)}(X_{n+1})\} \tag{6}$$

*for the true $S_{n+1}$ given a realization of $Y_{n+1} = y$ satisfies the desired coverage guarantee under all distribution shifts $f \in \mathcal{F}$, i.e.,*

$$P_f(Y_{n+1} \in C(X_{n+1})) \geq 1 - \alpha. \tag{7}$$

Building upon Lemma 1, we derive our first main result in Thm. 1. We define the finite-dimensional function class of interest as $\mathcal{F} := \{\theta \frac{\pi(a + \Delta_A | x)}{\pi(a | x)} \mid \theta \in \mathbb{R}^+\}$. It is easy to verify that all $f \in \mathcal{F}$ represent the desired propensity shift to $\tilde{\pi}(a \mid x) = \pi(a + \Delta_A \mid x)$ as defined in Eq. 2.

---

[4]We note that commonly used solvers, such as interior point solvers, are invariant to the data ordering.

**Theorem 1** (Conformal prediction intervals for known baseline policy). *Consider a new datapoint with $X_{n+1} = x_{n+1}$, $A_{n+1} = a_{n+1}$, and $A^*(X_{n+1}) = a^* = a_{n+1} + \Delta_A$. Let $\eta^S = \{\eta_{m+1}^S, \ldots, \eta_{n+1}^S\} \in \mathbb{R}^{n+1}$ be the optimal solution to*

$$\max_{\eta_i, i=m+1,\ldots,n+1} \min_{\theta > 0} \sum_{i=m+1}^{n} \eta_i \left( S_i - \theta \frac{\pi(a_i + \Delta_A \mid x_i)}{\pi(a_i \mid x_i)} \right) + \eta_{n+1} \left( S - \theta \frac{\pi(a^* \mid x_{n+1})}{\pi(a_{n+1} \mid x_{n+1})} \right)$$

*s.t.* $\quad -\alpha \leq \eta_i \leq 1 - \alpha, \quad \forall i = m+1, \ldots, n+1,$

$$(8)$$

*for an imputed unknown $S_{n+1} = S$. Furthermore, let $S^*$ be defined as the maximum $S$ s.t. $\eta_{n+1}^S < 1 - \alpha$. Then, the prediction interval*

$$C(x_{n+1}, a^*) := \{y \mid S_{n+1}(y) \leq S^*\} \tag{9}$$

*satisfies the desired coverage guarantee*

$$P(Y(A^*(X_{n+1})) \in C(X_{n+1}, A^*(X_{n+1}))) \geq 1 - \alpha. \tag{10}$$

*Proof.* We provide a full proof in Supplement D.2. Here, we briefly outline the underlying idea of the proof. First, we show that the function class $\mathcal{F}$ indeed satisfies Eq. equation 2 for the intervention $A^*(X) = A + \Delta_A$, and we then rewrite Eq. equation 5 as a convex optimization problem. Next, we exploit the strong duality property, we optimize over the corresponding dual problem to receive a dual prediction set with equal coverage probability. Finally, we derive $S^*$ from the dual prediction set to construct $C_{n+1}$ and prove the overall coverage guarantee. □

### 4.2 Scenario 2: Unknown treatment policy

If the underlying treatment policy is unknown, the only possible intervention is a hard intervention $a^*$. As described above, measuring the induced propensity shift is non-trivial due to two reasons: (i) The propensity model needs to be estimated, which introduces additional uncertainty affecting the validity of the intervals ($\rightarrow$ Challenge (b)). (ii) The density function corresponding to a hard intervention is given by the Dirac delta function

$$\delta_{a^*}(a) := \begin{cases} 0, & \text{for} \quad a \neq a^*, \\ \infty, & \text{for} \quad a = a^*, \end{cases} \tag{11}$$

which hinders a direct adaptation of Theorem 1 due to the inherent discontinuity of the improper function. Hence, to proceed, we make the following assumption on the propensity estimator.

**Assumption 1.** *The estimation error of the propensity function $\hat{\pi}(a \mid x)$ is bounded in the sense that, for all $i = 1, \ldots, n+1$, there exists $M > 0$ such that*

$$c_{a_i} := \frac{\hat{\pi}(a_i \mid x_i)}{\pi(a_i \mid x_i)} \in \left[ \frac{1}{M}, M \right]. \tag{12}$$

Under Assumption 1, the distribution shift induced by the intervention is then defined as

$$\tilde{\pi}(a \mid x) = \frac{\delta_{a^*}(a)}{\hat{\pi}(a \mid x)} \frac{\hat{\pi}(a \mid x)}{\pi(a \mid x)} \pi(a \mid x) = c_a \frac{\delta_{a^*}(a)}{\hat{\pi}(a \mid x)} \pi(a \mid x) = \frac{f(a, x)}{\mathbb{E}_P[f(A, X)]} \pi(a \mid x), \tag{13}$$

for a suitable function $f$. We further formulate $\delta_{a^*}(a)$ in terms of a Gaussian function as

$$\delta_{a^*}(a) = \lim_{\sigma \to 0} \frac{1}{\sqrt{2\pi}\sigma} \exp\left( -\frac{(a - a^*)^2}{2\sigma^2} \right). \tag{14}$$

This motivates the following lemma. Therein, we specify the class $\mathcal{F}$ of tilting functions $f$ that represent the distribution shift induced by the hard intervention $a^*$.

**Lemma 2.** *For $\sigma > 0$, we define*

$$f(a, x) := \frac{c_a}{\sqrt{2\pi}\sigma} \frac{\exp(-\frac{(a - a^*)^2}{2\sigma^2})}{\hat{\pi}(a \mid x)} \tag{15}$$

with $\mathbb{E}_P[f(A, X)] = 1$. *Furthermore, we define the finite-dimensional function class $\mathcal{F}$*

$$\mathcal{F} := \left\{ \frac{c_a}{\sqrt{2\pi}\sigma} \frac{\exp(-\frac{(a-a^*)^2}{2\sigma^2})}{\hat{\pi}(a \mid x)} \ \middle| \ 0 < \sigma, \frac{1}{M} \leq c_a \leq M \right\}. \tag{16}$$

*Then, $f(a, x) \in \mathcal{F}$ for all $c_a \in [\frac{1}{M}, M]$ and $\sigma \to 0$. As a result, the distribution shift*

$$\tilde{\pi}(a \mid x) = \lim_{\sigma \to 0} \frac{c_a}{\sqrt{2\pi}\sigma} \frac{\exp(-\frac{(a-a^*)^2}{\sigma^2})}{\hat{\pi}(a \mid x)} \pi(a \mid x) \tag{17}$$

*can be represented in terms of Eq. equation 2 through functions $f \in \mathcal{F}$.*

*Proof.* See Supplement D.1. $\qquad\square$

Following the motivation in scenario ①, we thus aim to estimate the $(1 - \alpha)$-quantile of the non-conformity scores under the distribution shift in Lemma 1. We can reformulate this problem as

$$\min_{\sigma > 0, \frac{1}{M} \leq c_a \leq M} \sum_{i=m+1}^{n+1} (1 - \alpha)u_i + \alpha v_i$$

$$\text{s.t.} \quad S_i - \frac{c_a}{\sqrt{2\pi}\sigma} \frac{\exp\left(-\frac{(a_i - a^*)^2}{2\sigma^2}\right)}{\hat{\pi}(a_i \mid x_i)} - u_i + v_i = 0, \quad \forall i = m+1 \dots, n+1, \tag{$P_S$}$$

$$u_i, v_i \geq 0, \quad \forall i = m+1 \dots, n+1$$

for the imputed score $S_{n+1} = S$. As the score is unknown, computing the CP interval would require solving equation $P_S$ for all $S \in \mathbb{R}$, yet which is computationally infeasible. As a remedy, we previously exploited properties of the dual optimization problem and the Lagrange multipliers of the convex problem in Theorem 1 to efficiently compute the CP intervals (see the proof of Theorem 1 in Supplement D). However, the present non-convex problem does not automatically allow for the same simplifications. Instead, we now present a remedy for efficient computation of the CP intervals in the following lemma. We prove Lemma 3 in Supplement D.

**Lemma 3.** *The problem equation $P_S$ is Type-I invex and satisfies the linear independence constraint qualification (LICQ).*

Lemma 3 allows us to derive properties of the present non-convex optimization problem in terms of the Karush-Kuhn-Tucker (KKT) conditions. For this, we note that the fulfillment of the LICQ serves as a sufficient regularity condition for the KKT to hold at any (local) optimum of equation $P_S$. Combined with the Type-I invexity of the objective function and the constraints, the KKT conditions are not only necessary but also sufficient for a global optimum. As a result, we can employ the KKT conditions at the optimal values[5] $\sigma^*$ and $c_a^*$ to derive coverage guarantees of our CP interval in a similar fashion as in Theorem 1. We thus arrive at the following theorem to provide CP intervals for the scenario with unknown propensity scores.

**Theorem 2** (Conformal prediction intervals for unknown propensity scores). *Let $u^S = \{u_{m+1}^S, \dots, u_{n+1}^S\}, v^S = \{v_{m+1}^S, \dots, v_{n+1}^S\} \in \mathbb{R}^{n-m}, \sigma^S, c_a^S \in \mathbb{R}$ be the optimal solution to*

$$\min_{\sigma > 0, \frac{1}{M} \leq c_a \leq M} \sum_{i=m+1}^{n+1} (1 - \alpha)u_i + \alpha v_i$$

$$\text{s.t.} \quad S_i - \frac{c_a}{\sqrt{2\pi}\sigma} \frac{\exp\left(-\frac{(a_i - a^*)^2}{2\sigma^2}\right)}{\hat{\pi}(a_i \mid x_i)} - u_i + v_i = 0, \quad \forall i = m+1 \dots, n+1, \tag{18}$$

$$u_i, v_i \geq 0, \quad \forall i = m+1 \dots, n+1.$$

*for an imputed unknown $S_{n+1} = S$. Let $S^*$ be defined as the maximum $S$ s.t. $v_{n+1}^S > 0$. Then, the interval*

$$C(X_{n+1}, a^*) := \{y \mid S_{n+1}(y) \leq S^*\} \tag{19}$$

*satisfies the desired coverage guarantee*

$$P(Y(a^*) \in C(X_{n+1}, a^*)) \geq 1 - \alpha. \tag{20}$$

---

[5]We provide an interpretation of the optimal values in Supplement F. Therein, we further discuss the implications of the proposed kernel smoothing of $\delta_{a^*}(a)$.

*Proof.* See Supplement D.3. □

In certain applications, it might be beneficial to fix $\sigma$ to a small value $\sigma_0$ to approximate $\delta_{a^*}(a)$ though a soft intervention and only construct the CP interval through optimizing over $c_a$. We present an alternative theorem for this case in Supplement A.

We now use Thm. 2 to present an algorithm for computing CP intervals of potential outcomes from continuous treatment variables under unknown propensities in Alg. 1. We present a similar algorithm for scenario ① with known propensities and discuss the computational complexity in Supplement B.

## 5 EXPERIMENTS

**Baselines:** As we have discussed above, there are *no* baselines that directly compute prediction intervals with finite-sample validity guarantees for potential outcomes of continuous treatments. Therefore, we compare our method against MC dropout (Gal & Ghahramani, 2016) and deep ensemble methods (Lakshminarayanan et al., 2017). Yet, we again emphasize that MC dropout is an ad hoc method with poor approximations of the posterior, which is known to give unfaithful intervals (Le Folgoc et al., 2021). Furthermore, we report the empirical coverage achieved by a vanilla CP (V-CP) method, i.e., CP which does not account for the distribution shift, and intervals from a Gaussian process regression (GP). By doing so, we (i) show the necessity of accounting for the distribution shift induced by the intervention, and (ii) consider a method that assesses the underlying aleatoric uncertainty.

**Implementation:** All methods are implemented with $\phi$ as a multi-layer perceptron (MLP) and an MC dropout regularization of rate 0.1. Crucially, we use the *identical* MLP for both our CP method and MC dropout. Hence, all performance gains must be attributed to the coverage guarantees of our conformal method. In the MC dropout baseline, the uncertainty intervals are computed via Monte Carlo sampling. In scenario ② with unknown propensity scores, we perform the conditional density estimation through conditional normalizing flows (Trippe & Turner, 2018). Details about our implementation and training are in Supplement F.

**Performance metrics:** We evaluate the methods in terms of whether the prediction intervals are *faithful* (e.g., Hess et al., 2024). That is, we compute whether the *empirical coverage* of the prediction intervals surpasses the threshold of $1 - \alpha$ for different significance levels $\alpha \in \{0.05, 0.1, 0.2\}$. Additionally, we report the width of the resulting intervals in Supplement G.

### 5.1 DATASETS

**Synthetic datasets:** We follow common practice in causal ML and evaluate our methods using synthetic datasets (e.g., Alaa et al., 2023; Jin et al., 2023). The reason is the fundamental problem of causal inference, because of which counterfactual outcomes are never observable in real-world datasets. Synthetic datasets enable us to access counterfactual outcomes and, thus, to benchmark methods in terms of whether the computed intervals are faithful. Additionally, we perform experiments on the semi-synthic TCGA dataset in Supplement C. We hereby show the applicability of our method to high-dimensional real-world data in a controlled environment.

We consider two synthetic datasets with different propensity scores and outcome functions. Dataset 1 uses a step-wise propensity function and a concave outcome function. Dataset 2 is more complex and uses a Gaussian propensity function and oscillating outcome functions. Both datasets contain a single discrete confounder, a continuous treatment, and a continuous outcome. By choosing low-dimensional datasets, we later render it possible to plot the treatment–response curves so that one can inspect the prediction intervals visually. (We later also show that our method scales to high-dimensional settings as part of the real-world dataset.) Details about the data-generating processes are in Supplement F.

**Medical dataset:** We demonstrate the applicability of our CP method to datasets from medicine by leveraging the MIMIC-III dataset (Johnson et al., 2016). MIMIC-III contains de-identified health records from patients admitted to critical care units at large tertiary care hospitals. Our goal is to predict patient outcomes in terms of blood pressure when treated with a different duration of mechanical ventilation. We use 8 confounders from medical practice (e.g., respiratory rate, hematocrit). Overall, we consider 14,719 patients, split into train (60%), validation (10%), calibration (20%), and test (10%) sets. Details are in Supplement F.

We now evaluate the faithfulness of our CP intervals. On each dataset, we analyze the performance of the prediction intervals in the presence of various soft interventions $\Delta_a \in \{1, 5, 10\}$ and hard interventions $a^* \in \{5x, 7x, 10x\}$ for each $X = x$. We average the empirical coverage across 50 runs with random seeds. The results are in Fig. 4 and Fig. 5. Additionally, we report the empirical coverage of the baselines V-CP and GP. (i) Across all distribution shifts and confidence levels, we observe that V-

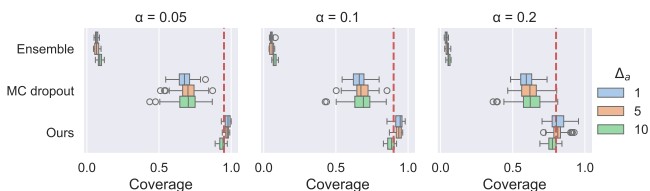

Figure 4: Comparison of *faithfulness* on dataset 1 across 50 runs. Larger values are better. For each $\alpha$, the plots show how often the empirical intervals contain the true outcome. Intervals should ideally yield a coverage of $1 - \alpha$ (red line).

CP cannot achieve any valid prediction interval. This is highly likely due to the good prediction performance of the underlying model (see Supplement G). Thus, V-CP intervals are extremely small (average width of 0.0003) and thus can never cover the true potential outcome after the intervention. (ii) The GP is only able to capture the true potential outcome in the prediction intervals for small distribution shifts ($\Delta = 1$) on dataset 2. However, the empirical coverage is extremely low: $\alpha = 0.05$: 0.125; $\alpha = 0.1$: 0.125; $\alpha = 0.2$: 0.0833. This result aligns with our expectations as the aleatoric uncertainty in our experiments is low. Therefore, the GP intervals will be small (average width of 0.1293) and barely provides valid intervals for the potential outcomes after the intervention.

## 5.2 RESULTS FOR SYNTHETIC DATASETS

We make the following observations. First, the intervals of our CP method comply with the targeted significance level $\alpha$ and, therefore, are faithful. Second, both MC dropout and the deep ensemble method have, in contrast, a considerably lower coverage, implying that the intervals are *not* faithful. This is in line with the literature, where MC dropout is found to produce poor approximations of the posterior (Le Folgoc et al., 2021). In

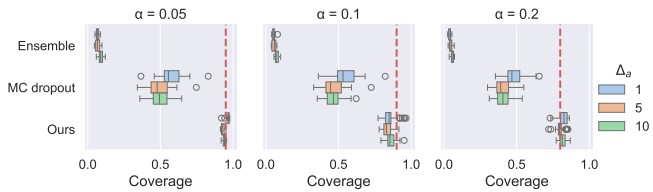

Figure 5: Comparison of *faithfulness* on dataset 2 across 50 runs. Larger values are better.

particular, the ensemble method is highly unfaithful. Thus, we will not consider this baseline in all of the following experiments. Third, our method has only a small variability in terms of empirical coverage, whereas the empirical coverage of MC dropout varies greatly. This corroborates the robustness of our method. Fourth, the results are consistent for both datasets. In sum, this demonstrates the effectiveness of our proposed CP method.

| | Coverage on dataset 1 | | | Coverage on dataset 2 | | |
|---|---|---|---|---|---|---|
| Intervention | $\alpha = 0.05$ | $\alpha = 0.1$ | $\alpha = 0.2$ | $\alpha = 0.05$ | $\alpha = 0.1$ | $\alpha = 0.2$ |
| $a^* = 7x$ | **1.00** / 0.19 | **0.90** / 0.13 | **0.83** / 0.11 | **1.00** / 0.02 | **0.94** / 0.02 | **0.85** / 0.02 |
| $a^* = 10x$ | **1.00** / 0.28 | **0.91** / 0.23 | **0.88** / 0.11 | **1.00** / 0.08 | **0.84** / 0.07 | **0.83** / 0.07 |

Table 1: Coverage of the intervals from our CP method / MC dropout for various hard interventions $a^*$ and significance levels $\alpha$. Intervals with coverage $\geq 1 - \alpha$ are considered faithful (bold numbers).

Table 1 presents the empirical coverage of the intervals from our CP method vs. MC dropout across different $\alpha$ and hard interventions $a^*$. We observe that our conformal prediction intervals are effective and achieve the intended coverage. In contrast, MC dropout does not provide faithful intervals. Our findings are again in line with the literature, where MC dropout is found to produce poor approximations of the posterior and thus might provide poor coverage (Le Folgoc et al., 2021). We present further results in Supplement G.

**Insights:** We plot the intervals across different significance levels $\alpha$ and covariates $X$ (see Fig. 6). This allows us to inspect the intervals visually. We observe that the intervals behave as expected: they become sharper with increasing significance level $\alpha$. We further see that our CP intervals are slightly wider (see details in Supplement G), yet this is intended because it ensures that the intervals are faithful. Our CP intervals (blue) generally include the true outcome. In contrast, the intervals from MC dropout (orange) often do *not* include the true outcome (e.g., see the bottom row Fig. 6) and are thus *not* faithful.

### 5.3 RESULTS FOR THE MIMIC DATASET

In Fig. 7, we compare the CP intervals of each two male and female patients of differing ages when treated with increasing duration of mechanical ventilation. Our intervals show higher uncertainty in treatment regions rarely included in the training data (high medium to high treatments). The intervals given by MC-dropout are narrower, which indicates lower coverage, thus confirming the effectiveness of our proposed method. This finding aligns with our observation from the synthetic datasets.

## 6 DISCUSSION

**Limitations:** As with any other method, our UQ method has limitations that offer opportunities for future research. Our method relies on the quality of the propensity estimator. Although we incorporate estimation errors in the construction of our intervals, poorly estimated propensities could potentially lead to wide prediction intervals. We acknowledge that our prediction intervals are conservative for the point intervention and for segments of the output space with limited calibration data, implying that a representative calibration dataset is essential for the performance of our method. As for all CP methods, the use of sample splitting may reduce data efficiency. Furthermore, we note that the optimization procedure can be computationally expensive for large CATE vectors.

**Broader impact:** Our method makes a significant step toward UQ for potential outcomes and, thus, toward *reliable* decision-making. We provided strong theoretical and empirical evidence that our prediction intervals are valid. To this end, our method fills an important demand for using causal ML in medical practice Feuerriegel et al. (2024) and other safety-critical applications with limited data.

**Conclusion:** We presented a novel conformal prediction method for potential outcomes of continuous treatments with finite-sample guarantees. Our method extends naturally to treatment effects. A key strength of our method is that the intervals are valid under distribution shifts introduced by the treatment assignment, even if the propensity score is unknown and has to be estimated.

Figure 6: Prediction intervals for multiple significance levels $\alpha$ for synthetic dataset 1 with intervention $\Delta = 5$.

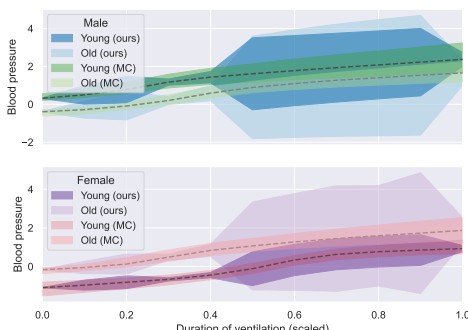

Figure 7: CP intervals for potential outcomes of increasing duration of mechanical ventilation for four exemplary patients.

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

# A  ADDITIONAL THEORETICAL RESULTS

## A.1  CALCULATING PREDICTION INTERVALS FOR FURTHER CAUSAL QUANTITIES AND DIFFERENCES

We presented a method for calculating conformal prediction intervals for potential outcomes of continuous treatments. In the following, we show how the intervals can be combined to yield valid prediction intervals for further causal quantities, such as the individual treatment effect (ITE) $\gamma_i$ of treatment $a$:

$$\gamma_i(a) := Y_i(a) - Y_i(0). \tag{21}$$

Here, we consider the setting in which the non-conformity score is chosen to be the absolute residual.

**Lemma 4.** *Let $S_a^*$ and $S_0^*$ denote the optimal imputed non-conformity scores $S_{n+1}$ for treatment $a$ and no treatment at significance level $1 - \frac{\alpha}{2}$ for $\alpha \in (0, 1)$, respectively. Furthermore, let*

$$C^+ := \phi(x_i, a) + S_a^* - \phi(x_i, 0) + S_0^*, \tag{22}$$

$$C^- := \phi(x_i, a) - S_a^* - \phi(x_i, 0) - S_0^*. \tag{23}$$

*Then the interval $C_\gamma(X_i, a) := [C^-, C^+]$ contains the ITE $\gamma_i$ with probability $1 - \alpha$.*

*Proof.* Let $\varepsilon_i(a)$ be the estimation error of the potential outcome, i.e.

$$\varepsilon_i(a) := Y_i(a) - \phi(x_i, a). \tag{24}$$

We can rewrite the coverage guarantee of the original conformal prediction intervals for the potential outcome $Y(a)$ as

$$P(Y_i(a) \in C(X_i, a)) = P(|\varepsilon_i(a)| \le S_a^*) \ge 1 - \frac{\alpha}{2}. \tag{25}$$

Now observe that

$$P(\gamma_i(a) \in C_\gamma(x_i, a)) = P((Y_i(a) - Y_i(0)) \in C_\gamma(x_i, a)) \tag{26}$$

$$= P((Y_i(a) \ge C^- + Y_i(0)) \wedge (Y_i(a) \le C^+ + Y_i(0))) \tag{27}$$

$$= P((\varepsilon_i(a) \ge \varepsilon_i(0) - (S_a^* + S_0^*)) \wedge (\varepsilon_i(a) \le \varepsilon_i(0) + (S_a^* + S_0^*))) \tag{28}$$

$$= P(|\varepsilon_i(a) - \varepsilon_i(0)| \le S_a^* + S_0^*) \tag{29}$$

$$\ge P(|\varepsilon_i(a)| + |\varepsilon_i(0)| \le S_a^* + S_0^*). \tag{30}$$

Thus, it follows directly that

$$P(\gamma_i(a) \in C_\gamma(X_i, a)) \ge 1 - \alpha. \tag{31}$$

$\square$

## A.2  ALTERNATIVE SCENARIO 2: FIXING AN APPROXIMATION OF $\delta_{a^*}(a)$

In Section 4.2, we formulated the unknown propensity shift in terms of

$$\delta_{a^*}(a) = \lim_{\sigma \to 0} \frac{1}{\sqrt{2\pi}\sigma} \exp\left(-\frac{(a - a^*)^2}{\sigma^2}\right) \tag{32}$$

and minimized over $\sigma$ and $c_a$ in Theorem 2 to construct the CP intervals. However, in certain applications, it might be beneficial to control the spread of the approximation of $\delta_{a^*}(a)$ through fixing $\sigma$ to a small value $\sigma_0$ and performing the soft intervention $\tilde{\pi}(a \mid x) = \frac{c_a}{\sqrt{2\pi}\sigma_0} \frac{\exp\left(-\frac{(a-a^*)^2}{\sigma_0^2}\right)}{\hat{\pi}(a|x)}$. In this case, the resulting optimization problem is a convex problem similar to Theorem 1. We present the alternative optimization problem below.

**Theorem 3** (Alternative for Theorem 2: Conformal prediction intervals for unknown propensity scores). *Let a new datapoint be given with $X_{n+1} = x_{n+1}$ and $A_{n+1} = a_{n+1}$. Let $\eta^S = \{\eta_1^S, \ldots, \eta_{n+1}^S\} \in \mathbb{R}^{n+1}$ be the optimal solution to*

$$\max_{\substack{\eta_i, \\ i=1,\ldots,n+1}} \min_{\frac{1}{M} \leq c_a \leq M} \sum_{i=1}^{n} \eta_i \left( S_i - \frac{c_a}{\sqrt{2\pi}\sigma_0} \frac{\exp\left(-\frac{(a_i - a^*)^2}{\sigma_0^2}\right)}{\hat{\pi}(a_i \mid x_i)} \right) + \eta_{n+1} \left( S - \frac{c_a}{\sqrt{2\pi}\sigma_0} \frac{1}{\hat{\pi}(a_i \mid x_{n+1})} \right)$$

$$\text{s.t.} \qquad -\alpha \leq \eta_i \leq 1 - \alpha, \quad \forall i = 1, \ldots, n+1, \tag{33}$$

*for an imputed unknown $S_{n+1} = S$. Furthermore, let $S^*$ be defined as the maximum $S$ s.t. $\eta_{n+1}^S < 1 - \alpha$. Then, the prediction interval*

$$C(x_{n+1}, a^*) := \{y \mid S_{n+1}(y) \leq S^*\} \tag{34}$$

*satisfies the desired coverage guarantee*

$$P(Y(a^*) \in C(X_{n+1}, a^*)) \geq 1 - \alpha, \tag{35}$$

*where with a slight abuse of notation $Y(a^*)$ denotes the potential outcome under the soft intervention $\tilde{\pi}$ above.*

*Proof.* The statement follows from Theorem 2. $\qquad\square$

### A.3 SOFT-INTERVENTIONS ON ESTIMATED PROPENSITIES

In the main paper, we presented algorithms for constructing prediction intervals for soft interventions if the propensity function is known and hard interventions if it is unknown. These are arguably the most common scenarios in practice. However, in some cases, one might also be interested in the effect of soft interventions on estimated propensity scores (e.g., Marmarelis, Myrl G., Morstater, Fred et al., 2024). Therefore, we present an alternative theorem for calculating conformal prediction intervals under soft interventions with estimated propensity scores below.

**Theorem 4** (Conformal prediction intervals for soft interventions with unknown propensity scores). *Let a new datapoint be given with $X_{n+1} = x_{n+1}$ and $A_{n+1} = a_{n+1}$. Furthermore, let $\hat{\pi}$ denote the estimated propensity score with estimation error bounded by $[\frac{1}{M}, M]$, $M > 0$. The soft intervention is represented by the shift given through $\Delta \in \mathbb{R}$. Let $\eta^S = \{\eta_1^S, \ldots, \eta_{n+1}^S\} \in \mathbb{R}^{n+1}$ be the optimal solution to*

$$\max_{\substack{\eta_i, \\ i=1,\ldots,n+1}} \min_{\frac{1}{M} \leq c_a \leq M} \sum_{i=1}^{n} \eta_i \left( S_i - \frac{c_a \hat{\pi}(a_i + \Delta \mid x_i)}{\hat{\pi}(a_i \mid x_i)} \right) + \eta_{n+1} \left( S - \frac{c_a \hat{\pi}(a_{n+1} + \Delta \mid x_{n+1})}{\hat{\pi}(a_{n+1} \mid x_{n+1})} \right)$$

$$\text{s.t.} \qquad -\alpha \leq \eta_i \leq 1 - \alpha, \quad \forall i = 1, \ldots, n+1, \tag{36}$$

*for an imputed unknown $S_{n+1} = S$. Furthermore, let $S^*$ be defined as the maximum $S$ s.t. $\eta_{n+1}^S < 1 - \alpha$. Then, the prediction interval*

$$C(x_{n+1}, a^*) := \{y \mid S_{n+1}(y) \leq S^*\} \tag{37}$$

*satisfies the desired coverage guarantee*

$$P(Y(a^*) \in C(X_{n+1}, a^*)) \geq 1 - \alpha, \tag{38}$$

*where with a slight abuse of notation $Y(a^*)$ denotes the potential outcome under the soft intervention represented by $\Delta$.*

*Proof.* The statement follows from Theorem 1 and Theorem 2. $\qquad\square$

# B  ALGORITHM

We now use Thm. 2 to present an algorithm for computing CP intervals of potential outcomes from continuous treatment variables under unknown propensities in Alg. 1. We present a similar algorithm for scenario ① with known propensities and discuss the computational complexity in below.

We make the following comments: In our algorithm, an optimization solver is used to calculate $v_{n+1}$ according to Theorem 2. The specific choice of the solver is left to the user. In our experiments in Section 5, we perform the optimization via interior point methods. Further, the overall goal of our algorithm is to find the optimal imputed non-conformity score $S^*$ such that the coverage guarantees hold. It can be implemented through suitable iterative search algorithms.

---

**Algorithm 1:** Algorithm for computing CP intervals of potential outcomes of continuous interventions under unknown propensities.

---

**Input:** Calibration data $(X_i, A_i, Y_i)_{i \in \{m+1,...,n\}}$, new sample $X_{n+1}$ and intervention $a^*$, significance level $\alpha$, prediction model $\phi$, propensity estimator $\hat{\pi}$, assumed error bound $M$, error tolerance $\varepsilon$, optimization solver

**Output:** CP interval $C_{n+1}$ for a new test sample

1 $S_{\text{up}} \leftarrow \max\{\max_{i=m+1,...,n} S_i, 1\}$; $S_{\text{low}} \leftarrow \min\{\min_{i=m+1,...,n} S_i, -1\}$;

    /* Calculate $v_{n+1}^{\text{up}}$, $v_{n+1}^{low}$                                       */

2 $v_{n+1}^{\text{up}} \leftarrow \text{solver}(\phi, \hat{\pi}, (X_i, A_i, Y_i)_{i \in \{m+1,...,n\}}, X_{n+1}, a^*, \alpha, M, S_{\text{up}})$;

3 $v_{n+1}^{\text{low}} \leftarrow \text{solver}((X_i, A_i, Y_i)_{i \in \{m+1,...,n\}}, X_{n+1}, a^*, \alpha, M, S_{\text{low}})$;

    /* Iterative search for $S^*$                                              */

4 **while** $v_{n+1}^{up} > 0$ **do**

5     |    $S_{\text{up}} \leftarrow 2S_{up}$;

6     |    $v_{n+1}^{\text{up}} \leftarrow \text{solver}(\phi, \hat{\pi}, (X_i, A_i, Y_i)_{i \in \{m+1,...,n\}}, X_{n+1}, a^*, \alpha, M, S_{\text{up}})$;

7 **end**

8 **while** $v_{n+1}^{\text{low}} = 0$ **do**

9     |    $S_{\text{low}} \leftarrow 0.5 S_{\text{low}}$;

10     |    $v_{n+1}^{\text{low}} \leftarrow \text{solver}(\phi, \hat{\pi}, (X_i, A_i, Y_i)_{i \in \{m+1,...,n\}}, X_{n+1}, a^*, \alpha, M, S_{\text{low}})$;

11 **end**

12 $S^* \leftarrow \frac{S_{\text{up}} + S_{\text{low}}}{2}$;

13 **while** $S_{\text{up}} - S_{\text{low}} > \varepsilon$ **do**

14     |    $v_{n+1}^{S^*} \leftarrow \text{solver}(\phi, \hat{\pi}, (X_i, A_i, Y_i)_{i \in \{m+1,...,n\}}, X_{n+1}, a^*, \alpha, M, S^*)$;

15     |    **if** $v_{n+1}^{S^*} > 0$ **then**

16     |      |    $S_{\text{low}} \leftarrow \frac{S_{\text{up}} + S_{\text{low}}}{2}$;

17     |    **end**

18     |    **else**

19     |      |    $S_v \leftarrow \frac{S_{\text{up}} + S_{\text{low}}}{2}$;

20     |    **end**

21     |    $S^* \leftarrow \frac{S_{\text{up}} + S_{\text{low}}}{2}$;

22 **end**

    /* Compute $C(X_{n+1}, a^*)$                                           */

23 **return** $C(X_{n+1}, a^*) = \{y \mid S_{n+1}(y) \leq S^*\}$

---

Below, we state a second algorithm that is applicable if the propensity score is known. In this case, a convex optimization solver can be used.

**Computational complexity:** The complexity of running our algorithms depends heavily on the employed optimization solver with complexity $\sigma_s(n_c)$ (e.g., polynomial complexity for suitable convex solvers) and the size of the calibration dataset $n_c$. This might become costly for large-scale calibration datasets in practice. The outer algorithm has a time complexity of at most $O(\log(\frac{S_{up} - S_{low}}{\varepsilon}) + 1)$. Overall, our algorithm has a fixed complexity of $O(\log(\frac{S_{up} - S_{low}}{\varepsilon}) + \sigma_s(n_c))$. The complexity of deriving intervals through MC-dropout or ensemble methods depends, however, on the number of MC samples or models, respectively. The latter thus scales with the precision of the intervals, which might be difficult to control. In our work, we use optimization as a tool to provide conformal prediction intervals. Future research should focus on developing more efficient optimization algorithms for this task.

**Algorithm 2:** Algorithm for computing CP intervals of potential outcomes of continuous interventions under known propensities.

---

**Input:** Calibration data $(X_i, A_i, Y_i)_{i \in \{m+1,\ldots,n\}}$, new sample $X_{n+1}$ and soft intervention $A^*(X_{n+1})$, significance level $\alpha$, prediction model $\phi$, error tolerance $\varepsilon$, optimization solver

**Output:** CP interval $C_{n+1}$ for a new test sample

1   $S_{\text{up}} \leftarrow \max\{\max_{i=m+1,\ldots,n} S_i, 1\}$; $S_{\text{low}} \leftarrow \min\{\min_{i=m+1,\ldots,n} S_i, -1\}$;

    /* Calculate $\eta_{n+1}^{\text{up}}$, $\eta_{n+1}^{low}$, where $\eta$ is the optimal solution to Eq. (8). */

2   $\eta_{n+1}^{\text{up}} \leftarrow \text{solver}(\phi, \hat{\pi}, (X_i, A_i, Y_i)_{i \in \{m+1,\ldots,n\}}, X_{n+1}, A^*(X_{n+1}), \alpha, S_{\text{up}})$;

3   $\eta_{n+1}^{\text{low}} \leftarrow \text{solver}((X_i, A_i, Y_i)_{i \in \{m+1,\ldots,n\}}, X_{n+1}, A^*(X_{n+1}), \alpha, S_{\text{low}})$;

    /* Iterative search for $S^*$                              */

4   **while** $\eta_{n+1}^{up} < 1 - \alpha$ **do**

5      |   $S_{\text{up}} \leftarrow 2 S_{up}$;

6      |   $\eta_{n+1}^{\text{up}} \leftarrow \text{solver}(\phi, \hat{\pi}, (X_i, A_i, Y_i)_{i \in \{m+1,\ldots,n\}}, X_{n+1}, A^*(X_{n+1}), \alpha, S_{\text{up}})$;

7   **end**

8   **while** $v_{n+1}^{\text{low}} >= 1 - \alpha$ **do**

9      |   $S_{\text{low}} \leftarrow 0.5 S_{\text{low}}$;

10     |   $\eta_{n+1}^{\text{low}} \leftarrow \text{solver}(\phi, \hat{\pi}, (X_i, A_i, Y_i)_{i \in \{m+1,\ldots,n\}}, X_{n+1}, A^*(X_{n+1}), \alpha, S_{\text{low}})$;

11   **end**

12   $S^* \leftarrow \frac{S_{\text{up}} + S_{\text{low}}}{2}$;

13   **while** $S_{\text{up}} - S_{\text{low}} > \varepsilon$ **do**

14     |   $\eta_{n+1}^{S^*} \leftarrow \text{solver}(\phi, \hat{\pi}, (X_i, A_i, Y_i)_{i \in \{m+1,\ldots,n\}}, X_{n+1}, a^*, \alpha, S^*)$;

15     |   **if** $\eta_{n+1}^{S^*} < 1 - \alpha$ **then**

16     |      |   $S_{\text{low}} \leftarrow \frac{S_{\text{up}} + S_{\text{low}}}{2}$;

17     |   **end**

18     |   **else**

19     |      |   $S_{up} \leftarrow \frac{S_{\text{up}} + S_{\text{low}}}{2}$;

20     |   **end**

21     |   $S^* \leftarrow \frac{S_{\text{up}} + S_{\text{low}}}{2}$;

22   **end**

    /* Compute $C(X_{n+1}, A^*(X_{n+1}))$                        */

23   **return** $C(X_{n+1}, A^*(X_{n+1})) = \{y \mid S_{n+1}(y) \le S^*\}$

## C  SEMI-SYNTHETIC EXPERIMENTS

To underline the effectiveness of our method, we perform additional experiments on the semi-synthetic TCGA dataset. The Cancer Genome Atlas (TCGA) dataset (Weinstein et al., 2013) consists of a comprehensive and diverse collection of gene expression data. The data was collected from patients with different cancer types. In our experiment, we consider the gene expression measurements of the 4,000 genes with the highest variability which we employ as our features $X$. The study cohort of consisted of a total of 9659 patients. We model a continuous treatment based on the sum of the 10 covariates with the highest variance and assign a treatment effect which is constant in the sum of the covariates.

As in the main paper, we construct CP intervals for different interventions and confidence levels $\alpha$. We state the empirical coverage of our method in Table 2 below. The prediction performance of the trained model on the hold-out test dataset is reported. We find that our method is highly effective.

| | Confidence level | | |
|---|---|---|---|
| Intervention | $\alpha = 0.05$ | $\alpha = 0.1$ | $\alpha = 0.2$ |
| $\Delta = 0.5$ | 0.9680 (0.0324) | 0.8920 (0.0391) | 0.8040 (0.0741) |
| $\Delta = 1.0$ | 0.9733 (0.0377) | 0.9500 (0.0500) | 0.7667 (0.0618) |
| $\Delta = 1.5$ | 0.9400 (0.0438) | 0.8920 (0.0699) | 0.8200 (0.0619) |

Table 2: Coverage of the intervals from our CP method on the TCGA dataset. We report the mean followed by the standard deviation in apprentices.

We observe that our method consistently achieves the desired coverage. To evaluate the usefulness of the intervals, we also report the interval width in Table 3 below. The range of the outcomes was 2.0.

| | Confidence level | | |
|---|---|---|---|
| Intervention | $\alpha = 0.05$ | $\alpha = 0.1$ | $\alpha = 0.2$ |
| $\Delta = 0.5$ | 0.1003 (0.0331) | 0.0843 (0.0252) | 0.0683 (0.0169) |
| $\Delta = 1.0$ | 0.1017 (0.0420) | 0.0877 (0.0349) | 0.0642 (0.0200) |
| $\Delta = 1.5$ | 0.0930 (0.0272) | 0.0822 (0.0260) | 0.0674 (0.0180) |

Table 3: Width of the intervals from our CP method on the TCGA dataset. We report the mean followed by the standard deviation in apprentices.

# D PROOFS

## D.1 PROOFS OF THE SUPPORTING LEMMAS

In the following, we prove Lemma 2 and Lemma 3 from our main paper.

**Proof of Lemma 2**   Recall the definition of the hard intervention

$$\tilde{\pi}(a \mid x) = \delta_{a^*}(a) = \frac{\delta_{a^*}(a)}{\hat{\pi}(a \mid x)} \frac{\hat{\pi}(a \mid x)}{\pi(a \mid x)} \pi(a \mid x), \tag{39}$$

where

$$\delta_{a^*}(a) = \lim_{\sigma \to 0} \frac{1}{\sqrt{2\pi}\sigma} \exp\left(-\frac{(a-a^*)^2}{\sigma^2}\right). \tag{40}$$

Under Assumption 1, we have

$$\frac{\hat{\pi}(a \mid x)}{\pi(a \mid x)} =: c_a \in [\frac{1}{M}, M] \tag{41}$$

for some $M > 0$ and all $a, x$. Then

$$\lim_{\sigma \to 0} \frac{1}{\sqrt{2\pi}\sigma} \frac{\exp\left(-\frac{(a-a^*)^2}{\sigma^2}\right)}{\hat{\pi}(a \mid x)} \frac{1}{M} \le \pi(a^*|x) \le \lim_{\sigma \to 0} \frac{1}{\sqrt{2\pi}\sigma} \frac{\exp\left(-\frac{(a-a^*)^2}{\sigma^2}\right)}{\hat{\pi}(a \mid x)} M. \tag{42}$$

Therefore, the distribution shift induced by the hard intervention can be represented as

$$f(a, x) = \lim_{\sigma \to 0} \frac{c_a}{\sqrt{2\pi}\sigma} \frac{\exp\left(-\frac{(a-a^*)^2}{\sigma^2}\right)}{\hat{\pi}(a \mid x)} \in \mathcal{F} := \left\{ \frac{c_a}{\sqrt{2\pi}\sigma} \frac{\exp\left(-\frac{(a-a^*)^2}{\sigma^2}\right)}{\hat{\pi}(a \mid x)} \,\middle|\, 0 < \sigma, c_a \in [\frac{1}{M}, M] \right\}. \tag{43}$$

$\square$

**Proof of Lemma 3**   We first prove that the problem equation $P_S$ fulfills the linear independence constraint qualifications. For all $i = m + 1, \ldots, n + 1$, we denote the constraints of problem equation $P_S$ as

$$h_i(u, v, c_a, \sigma) := S_i - u_i + v_i - \frac{c_a}{\sqrt{2\pi}\sigma} \frac{\exp(-\frac{(a_i-a^*)^2}{2\sigma})}{\hat{\pi}(a_i, x_i)}. \tag{44}$$

The gradient of $h_i$ is given by

$$\nabla h_i(u, v, c_a, \sigma) = \begin{bmatrix} \frac{\partial h_i}{\partial u_{m+1}}(u, v, c_a, \sigma) \\ \frac{\partial h_i}{\partial v_{m+1}}(u, v, c_a, \sigma) \\ \vdots \\ \frac{\partial h_i}{\partial u_i}(u, v, c_a, \sigma) \\ \frac{\partial h_i}{\partial v_i}(u, v, c_a, \sigma) \\ \vdots \\ \frac{\partial h_i}{\partial c_a}(u, v, c_a, \sigma) \\ \frac{\partial h_i}{\partial \sigma}(u, v, c_a, \sigma) \end{bmatrix} = \begin{bmatrix} 0 \\ 0 \\ \vdots \\ -1 \\ 1 \\ \vdots \\ -\frac{1}{\sqrt{2\pi}\sigma}\frac{\exp(-\frac{(a_i-a^*)^2}{2\sigma})}{\hat{\pi}(a_i, x_i)} \\ \frac{\exp(-\frac{(a_i-a^*)^2}{2\sigma})}{\hat{\pi}(a_i, x_i)}\left(\frac{c_a}{\sqrt{2\pi}\sigma^2} - \frac{c_a(a_i-a^*)^2}{2\sqrt{2\pi}\sigma^4}\right) \end{bmatrix}. \tag{45}$$

Therefore, with $\nabla h := (\nabla h_{m+1}, \ldots, \nabla h_{n+1})$ and $\lambda \in \mathbb{R}^{n+1}$, we obtain

$$\nabla h \cdot \lambda = 0 \iff \lambda = 0 \in \mathbb{R}^{n+1}. \tag{46}$$

As a result, the constraints are linearly independent. This property suffices for the KKT conditions to hold at any (local) optimum of equation $P_S$. To furthermore show that the KKT conditions are also sufficient for a global optimum, we show that equation $P_S$ is Type-I invex. An optimization problem with objective function $f(x)$ and constraints $g(x) <= 0$ with $x \in \mathbb{R}^{n+1}$ is Type-I invex at $x_0$, if there exists $\nu(x, x_0) \in \mathbb{R}^{n+1}$, such that

$$f(x) - f(x_0) \geq \nu(x, x_0)^T \nabla f(x_0), \tag{47}$$

and

$$-g(x_0) \geq \nu(x, x_0)^T \nabla g(x_0) \tag{48}$$

(Hanson & Mond, 1987). In problem equation $P_S$, the gradients of the objective function and of each constraint $h_i$ for all $i, j = m+1, \ldots, n+1$ at $x_0$ are given by

$$\frac{\partial \mathrm{obj}(u_0, v_0, c_{a_0}, \sigma_0)}{\partial u_i} = 1 - \alpha, \quad \frac{\partial \mathrm{obj}(u_0, v_0, c_{a_0}, \sigma_0)}{\partial v_i} = \alpha, \tag{49}$$

$$\frac{\partial \mathrm{obj}(u_0, v_0, c_{a_0}, \sigma_0)}{\partial c_a} = \frac{\partial \mathrm{obj}(u_0, v_0, c_{a_0}, \sigma_0)}{\partial \sigma} = 0 \tag{50}$$

$\forall i = m+1, \ldots, n+1$ and

$$\frac{\partial h_i}{\partial u_j}\Big|_{u_0, v_0, c_{a_0}, \sigma_0} = \begin{cases} -1, & \text{for } i = j, \\ 0, & \text{else,} \end{cases} \quad \frac{\partial h_i}{\partial v_j}\Big|_{u_0, v_0, c_{a_0}, \sigma_0} = \begin{cases} 1, & \text{for } i = j, \\ 0, & \text{else,} \end{cases} \tag{51}$$

$$\frac{\partial h_i}{\partial c_a}\Big|_{u_0, v_0, c_{a_0}, \sigma_0} = -\frac{1}{\sqrt{2\pi}\sigma_0} \frac{\exp\left(-\frac{(a_i - a^*)^2}{2\sigma_0}\right)}{\hat{\pi}(a_i, x_i)}, \tag{52}$$

$$\frac{\partial h_i}{\partial \sigma}\Big|_{u_0, v_0, c_{a_0}, \sigma_0} = \frac{\exp\left(-\frac{(a_i - a^*)^2}{2\sigma_0}\right)}{\hat{\pi}(a_i, x_i)} \left(\frac{c_{a_0}}{\sqrt{2\pi}\sigma_0^2} - \frac{c_{a_0}(a_i - a^*)^2}{2\sqrt{2\pi}\sigma_0^4}\right). \tag{53}$$

For

$$\eta((u, v, c_a, \sigma), (u_0, v_0, c_{a_0}, \sigma_0)) := (-u_{0_1}, \ldots, -u_{0_{n+1}}, -v_{0_1}, \ldots, -v_{0_{n+1}}, -c_{a_0}, 0)^T, \tag{54}$$

the definition of Type-I invexity holds for equation $P_S$. Thus, the KKT conditions are also sufficient for a global optimum. $\qquad\square$

### D.2 PROOF OF THEOREM 1

We prove Theorem 1 in three steps: (i) We show that function class $\mathcal{F} := \{\theta \frac{\pi(a + \Delta_A | x)}{\pi(a | x)} \mid \theta \in \mathbb{R}^+\}$ indeed satisfies Eq. equation 2 for the intervention $a^* = a + \Delta_A$ and rewrite Eq. equation 5 as a convex optimization problem. (ii) We retrieve the corresponding dual problem, derive a dual prediction set, and show the equality of the coverage guarantee of the dual and the primal prediction sets. (iii) We derive $S^*$ from the dual prediction set to construct $C_{n+1}$ and prove the overall coverage guarantee. For further theoretical background on the idea of the proof, we refer to Gibbs et al. (2023).

**Justification of the distribution shift** Observe that $\mathbb{E}[f(A, X)] = \theta$ for all $f \in \mathcal{F} := \{\theta \frac{\pi(a + \Delta_A | x)}{\pi(a | x)} \mid \theta \in \mathbb{R}^+\}$. Therefore, Eq. equation 2 simplifies to

$$\tilde{\pi}(a, x) = \frac{\pi(a + \Delta_A \mid x)}{\pi(a \mid x)} \pi(a \mid x) = \pi(a + \Delta_A \mid x). \tag{55}$$

Thus, $\mathcal{F}$ satisfies the propensity shift from Eq. equation 2 for the soft intervention $a^* = a + \Delta_A$. Following Lemma 1, we thus aim to find

$$\hat{q}_S := \arg\min_{\theta > 0} \frac{1}{n - m} \left( \sum_{i=m+1}^{n} l_\alpha(\theta \frac{\pi(a_i + \Delta_A \mid x_i)}{\pi(a_i \mid x_i)}, S_i) + l_\alpha(\theta \frac{\pi(a^* \mid x_{n+1})}{\pi(a_{n+1} \mid x_{n+1})}, S) \right). \tag{56}$$

**Dual problem formulation**   First, we rewrite the primal problem as

$$\min_{\theta > 0} \quad \sum_{i=m+1}^{n+1} (1-\alpha)u_i + \alpha v_i$$

$$\text{s.t.} \quad S_i - \theta \frac{\pi(a_i + \Delta_A \mid x_i)}{\pi(a_i \mid x_i)}) - u_i + v_i = 0, \quad \forall i = m+1 \ldots, n+1, S_{n+1} = S$$

$$u_i, v_i \geq 0, \quad \forall i = m+1 \ldots, n+1. \tag{57}$$

For a reference, see (Gibbs et al., 2023). The Lagrangian of the primal problem states

$$\mathcal{L} = \sum_{i=m+1}^{n+1} (1-\alpha)u_i + \alpha v_i + \sum_{i=m+1}^{n+1} \eta_i \left( S_i - \theta \frac{\pi(a_i + \Delta_A \mid x_i)}{\pi(a_i \mid x_i)} - u_i + v_i \right) - \sum_{i=m+1}^{n+1} (\gamma_{1_i} u_i + \gamma_{2_i} v_i). \tag{58}$$

Setting derivative of $\mathcal{L}$ w.r.t. $u_i$ and $v_i$ to 0 results in

$$\frac{\partial \mathcal{L}}{\partial u_i} = (1-\alpha) - \eta_i - \gamma_{1_i} \overset{!}{=} 0, \quad \forall i = m+1 \ldots, n+1 \tag{59}$$

$$\frac{\partial \mathcal{L}}{\partial v_i} = (1-\alpha) - \eta_i - \gamma_{2_i} \overset{!}{=} 0, \quad \forall i = m+1 \ldots, n+1. \tag{60}$$

Since $\gamma_{1_i}, \gamma_{2_i} \geq 0 \, \forall i$, it follows for all $i = m+1, \ldots, n+1$ that

$$(1-\alpha) - \eta_i \geq 0 \quad \text{and} \quad \alpha - \eta_i \geq 0 \quad \Rightarrow -\alpha \leq \eta_i \leq 1 - \alpha. \tag{61}$$

Therefore, the dual problem is formulated as

$$\max_{\eta_i, i=m+1,\ldots,n+1} \min_{\theta > 0} \quad \sum_{i=m+1}^{n} \eta_i \left( S_i - \theta \frac{\pi(a_i + \Delta_A \mid x_i)}{\pi(a_i \mid x_i)} \right) + \eta_{n+1} \left( S - \theta \frac{\pi(a^* \mid x_{n+1})}{\pi(a_{n+1} \mid x_{n+1})} \right)$$

$$\text{s.t.} \quad -\alpha \leq \eta_i \leq 1 - \alpha, \quad \forall i = m+1, \ldots, n+1. \tag{62}$$

**Coverage guarantee**   Recall from Lemma 1 that, for

$$\hat{q}_{S_{n+1}}(y) = \arg \min_{\theta > 0} \frac{1}{n-m} \Big( \sum_{i=m+1}^{n} l_\alpha(\theta \frac{\pi(a_i + \Delta_A \mid x_i)}{\pi(a_i \mid x_i)}, S_i) + l_\alpha(\theta \frac{\pi(a^* \mid x_{n+1})}{\pi(a_{n+1} \mid x_{n+1})}, S_{n+1}(y))\Big), \tag{63}$$

we can construct $C_{n+1} = \{y \mid S_{n+1}(y) \leq \hat{q}_{S_{n+1}}(y)\}$ to achieve the desired coverage guarantee

$$P_f(Y(a^*) \in C(X_{n+1}, A^*(X_{n+1}))) \geq 1 - \alpha. \tag{64}$$

It is infeasible to calculate $\hat{q}_{S_{n+1}}(y)$ directly. Therefore, we optimize the dual problem to receive

$$C(X_{n+1}, a^*) := \{y \mid S_{n+1}(y) \leq S^*\} \tag{65}$$

with $S^*$ the maximum $S$, s.t. for $\eta_{n+1}^S$ maximizing the dual problem, $\eta_{n+1}^S < 1 - \alpha$. Hence, it is left to show that replacing $\hat{q}_{S_{n+1}}(y)$ by $S^*$ in $C$ does not change to coverage guarantee.

To do so, we fix some $\theta > 0$ to obtain a specific $f(a, x) := \theta \frac{\pi(a^* \mid x)}{\pi(a \mid x)}$. Let $\hat{g}(a, x) \in \mathcal{F}$ denote the primal optimal solution. Recall the Lagrangian

$$\mathcal{L} = \sum_{i=m+1}^{n+1} (1-\alpha)u_i + \alpha v_i + \sum_{i=m+1}^{n+1} \eta_i(S_i - f(a_i, x_i) - u_i + v_i) - \sum_{i=m+1}^{n+1} (\gamma_{1_i} u_i + \gamma_{2_i} v_i). \tag{66}$$

Deriving wrt. $f$ yields the stationarity condition

$$0 \overset{!}{=} - \sum_{i=m+1}^{n+1} \eta_i^S f(a_i, x_i) \tag{67}$$

$$= - \sum_{S_i < \hat{g}(a_i, x_i)} \eta_i^S f(a_i, x_i) - \sum_{S_i > \hat{g}(a_i, x_i)} \eta_i^S f(a_i, x_i) - \sum_{S_i = \hat{g}(a_i, x_i)} \eta_i^S f(a_i, x_i). \tag{68}$$

The complementary slackness Karush-Kuhn-Tucker conditions yield

$$\eta_i^S \in \begin{cases} -\alpha, & \text{if } S_i < \hat{g}(a_i, x_i), \\ [-\alpha, 1-\alpha], & \text{if } S_i = \hat{g}(a_i, x_i), \\ 1-\alpha, & \text{if } S_i > \hat{g}(a_i, x_i). \end{cases} \tag{69}$$

Therefore, we can rewrite the equation from above as

$$0 = \sum_{S_i < \hat{g}(a_i, x_i)} \alpha f(a_i, x_i) - \sum_{S_i > \hat{g}(a_i, x_i)} (1-\alpha) f(a_i, x_i) - \sum_{S_i = \hat{g}(a_i, x_i)} \eta_i^S f(a_i, x_i) \tag{70}$$

$$= \sum_{\eta_i^S < 1-\alpha} \alpha f(a_i, x_i) - \sum_{\eta_i^S = 1-\alpha} (1-\alpha) f(a_i, x_i) - \sum_{\substack{\eta_i^S < 1-\alpha, \\ S_i = \hat{g}(a_i, x_i)}} (\alpha + \eta_i^S) f(a_i, x_i) \tag{71}$$

$$= \sum_{i=m+1}^{n+1} (\alpha - \mathbb{1}_{[\eta_i^S = 1-\alpha]}) f(a_i, x_i) - \sum_{\substack{\eta_i^S < 1-\alpha, \\ S_i = \hat{g}(a_i, x_i)}} (\alpha + \eta_i^S) f(a_i, x_i). \tag{72}$$

Before deriving the coverage guarantee from the stationarity condition, we state the following lemma to underline the definition of $S^*$.

**Lemma 5** (Gibbs et al. (2023))**.** *The mapping $S \mapsto \eta_{n+1}^S$ is non-decreasing in $S$ for all $\eta_{n+1}^S$ maximizing*

$$\max_{\eta_i, i=m+1,\ldots,n+1} \min_{g \in \mathcal{F}} \quad \sum_{i=1}^{n} \eta_i (S_i - g(a_i, x_i)) + \eta_{n+1}(S - g(a_{n+1}, x_{n+1}))$$

$$s.t. \qquad -\alpha \leq \eta_i \leq 1-\alpha, \quad \forall i = m+1, \ldots, n+1 \tag{73}$$

*for non-negative function classes $\mathcal{F}$.*

To prove the final coverage guarantee, we observe that

$$\mathbb{E}[f(a_{n+1}, x_{n+1})(\mathbb{1}_{[Y(a^*) \in C(X_{n+1}, a^*)]} - (1-\alpha))] \tag{74}$$

$$= \mathbb{E}[f(a_{n+1}, x_{n+1})(\alpha - \mathbb{1}_{[Y(a^*) \notin C(X_{n+1}, a^*)]})] \tag{75}$$

$$= \mathbb{E}[f(a_{n+1}, x_{n+1})(\alpha - \mathbb{1}_{[S(y) > S^*]})]. \tag{76}$$

With the definition of $S^*$ as the maximum optimizer $\eta_{n+1}^S$ with $\eta_{n+1}^S < 1-\alpha$ and Lemma 5, it follows that

$$\mathbb{E}[f(a_{n+1}, x_{n+1})(\alpha - \mathbb{1}_{[S(y) > S^*]})] = \mathbb{E}[(\alpha - \mathbb{1}_{[\eta_{n+1}^S = 1-\alpha]}) f(a_{n+1}, x_{n+1})] \tag{77}$$

and, by exchangeability of $(f(a_i, x_i), \hat{q}_S(a_i. x_i), S_i)$, that

$$\mathbb{E}[(\alpha - \mathbb{1}_{[\eta_i^S = 1-\alpha]}) f(a_i, x_i)] = \mathbb{E}\left[ \frac{1}{n-m} \sum_{i=m+1}^{n+1} (\alpha - \mathbb{1}_{[\eta_i^S = 1-\alpha]}) f(a_i, x_i) \right] \tag{78}$$

$$= \mathbb{E}\left[ \frac{1}{n-m} \sum_{\substack{\eta_i^S < 1-\alpha, \\ S_i = \hat{g}(a_i, x_i)}} (\alpha + \eta_i^S) f(a_i, x_i) \right]. \tag{79}$$

Since $f$ is positive and $\eta_i \in [-\alpha, 1-\alpha]$, it follows

$$\mathbb{E}[f(a_{n+1}, x_{n+1})(\mathbb{1}_{[Y(a^*) \in C(X_{n+1}, a^*)]} - (1-\alpha))] \geq 0 \tag{80}$$

and thus

$$P_f(Y(a^*) \in C_{n+1} C(X_{n+1}, A^*(X_{n+1}))) \geq 1-\alpha. \tag{81}$$

$\square$

### D.3 Proof of Theorem 2

We follow the same outline as in the proof of Theorem 1 in Section D.2. In Lemma 2, we motivated the functional class of distribution shifts. Therefore, it is left to prove the coverage guarantee of $C_{n+1}$.

Key to our proof is the following lemma.

**Lemma 6.** *The mapping $S \mapsto v_{n+1}^S$ is non-increasing in S for all $g^S(x,a)$ minimizing*

$$\min_{g \in \mathcal{F}} \quad \sum_{i=m+1}^{n+1} (1-\alpha)u_i + \alpha v_i \tag{82}$$

$$s.t. \quad S_i - g(x_i, a_i) - u_i + v_i = 0, \quad \forall i = m+1, \dots, n+1$$

*for non-negative function classes $\mathcal{F}$ and imputed $S_{n+1} = S$ stemming from a non-negative non-conformity score function (e.g., the residual of the prediction).*

*Proof.* Assume for contradiction that there exists $\tilde{S} > S$ such that $v_{n+1}^{\tilde{S}} > v_{n+1}^S$. Then

$$(\tilde{S} - S)(v_{n+1}^{\tilde{S}} - v_{n+1}^S) > 0. \tag{83}$$

We observe that

$$\tilde{S}(S - g^S(x_{n+1}, a_{n+1}) - u_{n+1}^S + v_{n+1}^S) = S(\tilde{S} - g^{\tilde{S}}(x_{n+1}, a_{n+1}) - u_{n+1}^{\tilde{S}} + v_{n+1}^{\tilde{S}}) = 0. \tag{84}$$

Reformulating the equation above yields

$$(\tilde{S} - S)(v_{n+1}^{\tilde{S}} - v_{n+1}^S) \tag{85}$$

$$= \tilde{S}u_{n+1}^S - Su_{n+1}^{\tilde{S}} + \tilde{S}g^S(x_{n+1}, a_{n+1}) - Sg^{\tilde{S}}(x_{n+1}, a_{n+1}) + \tilde{S}v_{n+1}^S - Sv_{n+1}^S \tag{86}$$

$$< S(u_{n+1}^S - u_{n+1}^{\tilde{S}} + g^S(x_{n+1}, a_{n+1}) - g^{\tilde{S}}(x_{n+1}, a_{n+1}) - (v_{n+1}^S - v_{n+1}^{\tilde{S}})) \tag{87}$$

$$= S(S - \tilde{S}). \tag{88}$$

This is equivalent to

$$(S - \tilde{S})(v_{n+1}^S - v_{n+1}^{\tilde{S}}) < S(S - \tilde{S}) \tag{89}$$

$$\iff \quad v_{n+1}^S - v_{n+1}^{\tilde{S}} > S \geq 0, \tag{90}$$

which contradicts the assumption that $v_{n+1}^{\tilde{S}} > v_{n+1}^S$. $\qquad\square$

**Coverage guarantees.** As in D.2, we fix some $\sigma > 0$ and $c_a \in [\frac{1}{M}, M]$ to obtain a specific $f(a,x) := \frac{c_a}{\sqrt{2\pi}\sigma} \frac{\exp\left(-\frac{(a_i - a^*)^2}{2\sigma^2}\right)}{\hat{\pi}(a_i | x_i)}$. We further denote $\hat{g}(a,x) \in \mathcal{F}$ the optimal solution given by the optimal values $\hat{\sigma}$ and $\hat{c}_a$.

With the definition of $S^*$ as the minimum $S$ such that $v_{n+1}^{S^*} = 0$ and Lemma 6, we now can state

$$\mathbb{E}[f(a_{n+1}, x_{n+1})(\mathbb{1}_{[Y(a^*) \in C_{n+1}]} - (1-\alpha))] = \mathbb{E}[f(a_{n+1}, x_{n+1})(\mathbb{1}_{[v_{n+1}^S > 0]} - (1-\alpha))] \tag{91}$$

$$= \mathbb{E}[f(a_{n+1}, x_{n+1})(\alpha - \mathbb{1}_{[v_{n+1}^S = 0]})] \tag{92}$$

and, by exchangeability of $(f(a_i, x_i), \hat{q}_S(a_i.x_i), S_i)$, that

$$\mathbb{E}[f(a_{n+1}, x_{n+1})(\alpha - \mathbb{1}_{[v_{n+1}^S = 0]})] = \mathbb{E}\left[\frac{1}{n-m} \sum_{i=m+1}^{n+1} f(a_{n+1}, x_{n+1})(\alpha - \mathbb{1}_{[v_{n+1}^S = 0]})\right] \tag{93}$$

$$= \frac{1}{n-m}\mathbb{E}\left[\sum_{v_i^S > 0} \alpha f(a_i, x_i) - \sum_{v_i^S = 0})(1-\alpha)f(a_i, x_i)\right] \tag{94}$$

$$= \frac{1}{n-m}\mathbb{E}\left[\sum_{S_i < \hat{g}(a_i, x_i)} \alpha f(a_i, x_i) - \sum_{S_i \geq \hat{g}(a_i, x_i)})(1-\alpha)f(a_i, x_i)\right]. \tag{95}$$

Deriving the Lagrangian above wrt. $f$ yields the stationarity condition

$$0 \overset{!}{=} \sum_{i=m+1}^{n+1} \eta_i^S f(a_i, x_i) \tag{96}$$

$$= \sum_{S_i < \hat{g}(a_i, x_i)} \eta_i^S f(a_i, x_i) + \sum_{S_i > \hat{g}(a_i, x_i)} \eta_i^S f(a_i, x_i) + \sum_{S_i = \hat{g}(a_i, x_i)} \eta_i^S f(a_i, x_i). \tag{97}$$

The complementary slackness Karush-Kuhn-Tucker conditions yield

$$\eta_i^S \in \begin{cases} -\alpha, & \text{if } S_i < \hat{g}(a_i, x_i), \\ [-\alpha, 1-\alpha], & \text{if } S_i = \hat{g}(a_i, x_i), \\ 1-\alpha, & \text{if } S_i > \hat{g}(a_i, x_i). \end{cases} \tag{98}$$

Therefore, we receive

$$\mathbb{E}[f(a_{n+1}, x_{n+1})(\alpha - \mathbb{1}_{[v_{n+1}^S = 0]})] = \frac{1}{n-m} \mathbb{E}\left[ \sum_{\eta_i^S < 1-\alpha} \alpha f(a_i, x_i) - \sum_{\eta_i^S = 1-\alpha} )(1-\alpha) f(a_i, x_i) \right] \tag{99}$$

$$= \frac{1}{n-m} \mathbb{E}\left[ \sum_{\substack{\eta_i^S < 1-\alpha, \\ S_i = \hat{g}(a_i, x_i)}} (\alpha + \eta_i^S) f(a_i, x_i) \right]. \tag{100}$$

Since $f$ is positive and $\eta_i \in [-\alpha, 1-\alpha]$, it follows

$$\mathbb{E}[f(a_{n+1}, x_{n+1})(\mathbb{1}_{[Y(a^*) \in C(X_{n+1}, a^*)]} - (1-\alpha))] \geq 0 \tag{101}$$

and thus

$$P_f(Y(a^*) \in C(X_{n+1}, a^*)) \geq 1 - \alpha. \tag{102}$$

$\square$

# E  ADDITIONAL BACKGROUND

## E.1  EXTENDED LITERATURE REVIEW

**Uncertainty quantification for causal quantities**

There exist various methods for uncertainty quantification of causal quantities. These are often based on Bayesian methods (e.g., Alaa & van der Schaar, 2017; Hess et al., 2024; Hill, 2011; Jesson et al., 2020). However, Bayesian methods require the specification of a prior distribution based on domain knowledge and are thus neither robust to model misspecification nor generalizable to model-agnostic machine learning models. Other methods only provide asymptotic guarantees (e.g., Jin et al., 2023; Jonkers et al., 2024). The strength of conformal prediction, however, is to provide finite-sample uncertainty guarantees.

In the following, we present related work on CP for causal quantities in more detail.

Recently, Alaa et al. (2023) provided predictive intervals for CATE meta-learners under the assumption of full knowledge of the propensity score. As an extension, Jonkers et al. (2024) proposed a Monte-Carlo sampling approach to receive less conservative intervals. Chen et al. (2024) provide prediction intervals for counterfactual outcomes. However, the proposed method requires access to additional interventional data and is thus not applicable to real-world applications on observational data. All methods are restricted to binary treatments.

Other works focus on prediction intervals for off-policy prediction (Taufiq et al., 2022; Zhang et al., 2023) and conformal sensitivity analysis (Yin et al., 2022), thus neglecting estimation errors arising from propensity or weight estimation or for randomized control trials (Kivaranovic et al., 2020). Wang et al. (2024) constructed intervals with treatment-conditional coverage of discrete treatments. Aiming for group-conditional coverage, Wang et al. (2024) adapted CP to cluster randomized trials. Nevertheless, the method only applies to a finite number of treatments and thus is not applicable to continuous treatments.

Lei & Candès (2021) consider the estimated propensity by incorporating the estimation error as a TV-distance term in the coverage guarantees. However, for large TV-distances (close to 1), the proposed method can only construct intervals with a very limited coverage $\alpha \in (0, 1 - TV)$. Hence, the method is not suitable for applications in medical practice. Our method, however, can also construct intervals with high coverage guarantees for high estimation errors. An increased error will widen the prediction intervals instead of reducing the coverage guarantee. We consider our approach more suitable for medical practice, as one can visually inspect the intervals and decide on the suitability of the task at hand.

Overall, no method can provide exact intervals for continuous treatments. Especially, no method considers the error arising from propensity estimation in the analysis.

**Conformal prediction under covariate shift**

Multiple works on CP with *marginal coverage* under distribution shifts between training and test data have been introduced in the literature (e.g., Cauchois et al., 2020; Fannjiang et al., 2022; Gendler et al., 2022; Ghosh et al., 2023; Gibbs & Candès, 2021; Gibbs et al., 2023; Guan, 2023; Lei & Candès, 2021; Podkopaev & Ramdas, 2021; Tibshirani et al., 2019; Yang et al., 2024). Our setting also involves a distribution shift due to the intervention on the treatment but differs from the latter in that the true distribution shift is unknown.

Gibbs et al. (2023) introduced an approach to derive CP intervals under unknown distribution shifts. It proves valid finite-sample prediction intervals for all distribution shifts in a finite-dimensional function class. However, the approach does **not** directly apply to causal inference settings. Nevertheless, our framework builds upon the work by Gibbs et al. (2023) in that we re-frame the proposed approach to apply to the distribution shift induced through the intervention in causal effect estimation. In this setting, the distribution shift is captured by the shift of the propensity function. Adapting Gibbs et al. (2023) to a causal inference setting requires carefully addressing the underlying challenges that come from computing CP intervals in a causal inference setting (e.g., propensity score estimation, hard/soft interventions), which we regard as our main novelty and which is of immediate practical relevance (e.g., in personalized medicine).

### E.2  A NOTE ON CHALLENGES AND DIFFICULTIES IN CP FOR CAUSAL EFFECTS OF CONTINUOUS TREATMENTS

Existing works on conformal prediction for binary or (low-dimensional) discrete treatment are commonly based on (a) weighted conformal prediction (Tibshirani et al., 2019) or (b) conformal prediction local coverage guarantees (Lei & Wasserman, 2014). The first approach provides marginal coverage under a distribution shift through reweighting. It requires computing the weights based on the probability of treatment $A = a$. However, for continuous treatments, this is always zero. Although applicable to binary or low-dimensional discrete treatments (e.g., Lei & Candès, 2021), this weighting approach cannot be extended similarly to continuous treatments. Furthermore, the propensity of a continuous treatment given by the Dirac delta function $\delta_a$ would require us to restrict the calibration to data samples of the specific treatment, which are extremely rare or even *might be missing*. Therefore, the calibration step cannot be employed in our setting. The second approach provides treatment group-conditional coverage. Although again possible for binary or low-dimensional treatments, this approach *does not apply to continuous treatment*s as no treatment groups can be defined. Instead, we propose a novel method for conformal predictions that circumvents the above problems and is carefully tailored to continuous treatments.

### E.3  CAUSAL EFFECTS OF CONTINUOUS TREATMENTS & KERNEL SMOOTHING

Causal inference becomes challenging with continuous treatments primarily due to the infinite number of potential outcomes per sample, from which only one outcome is observed. Continuous treatments thus result in causal effects that are generally represented by curves (called dose-response curves) (**?**). This is unlike binary treatment, where the causal effects are represented by a single discrete value.

For continuous treatments, the dose-response curves are typically assumed to fulfill some smoothness criterion (e.g., Patrick Schwab et al., 2020; Schweisthal et al., 2023). Hence, when estimating treatment effects, interpolation and kernel smoothing of the outcome function are commonly employed (e.g., Kennedy, 2019; Nagalapatti et al., 2024).

Underlying causal estimation with continuous treatments is the generalized propensity score (Imbens, 2000). It is defined as the conditional probability of receiving treatment $a^*$ given the covariates $X$ under the following regularity conditions: (i) For each $i$, $Y_i(a), x_i, A_i$ are defined on a common probability space; (ii) $A_i$ is continuously distributed with respect to the Lebesgue measure; and (iii) $Y_i = Y_i(A_i)$ is a well-defined random variable.

Approximating the density $\delta_{a^*}(a)$ of the hard intervention $a^*$ through a Gaussian kernel follows directly from the definition of $\delta_{a^*}(a)$ as the limit of such kernel. This is also common in the literature (e.g., Kallus & Zhou, 2018). Importantly, we note that we do not directly approximate the potential outcome $Y(a^*)$ (but only the propensity scores). Thus, we do not have a bias-variance trade-off of the estimated outcome. Due to the smoothness of the dose-response curve, it is now valid to employ observed samples within a treatment region of $a^*$ defined by $\sigma$ to construct the intervals. We note that the importance of the samples is weighted by the inverse distance of the sample to $a^*$ in treatment space. We give further intuition on the relationship between $\sigma$, the importance of observational samples, and the prediction interval width in the following.

### E.4  INTERPRETATION OF OPTIMAL PARAMETERS

To obtain CP intervals under an unknown distribution shift, we approximate the Dirac-delta distribution representing the hard intervention by a Gaussian function as

$$\delta_{a^*}(a) = \lim_{\sigma \to 0} \frac{1}{\sqrt{2\pi}\sigma} \exp\left(-\frac{(a - a^*)^2}{2\sigma^2}\right). \tag{103}$$

In Theorem 2, we thus optimize over $\sigma > 0$ and $c_a \in [\frac{1}{M}, M]$ to obtain the $(1 - \alpha)$-quantile of the distribution shift-calibrated non-conformity scores. The optimal parameter $\sigma^*$ represents a trade-off between the uncertainty in the prediction and the uncertainty in the interval construction: A small $\sigma^*$ resembles the propensity of the hard intervention best. Thus, with sufficient or even infinite data close to $a^*$, we could construct the narrowest CP interval. However, the smaller $\sigma$, the less data close to $a^*$ will be available to calculate the prediction interval in practice. As a result, many calibration

data samples will be strongly perturbed during the calculation, which increases the uncertainty and, thus, the interval size.

The parameter $c_a$ allows us to incorporate the estimation error in the propensity score. It represents a weighting of the propensity shift such that the $(1 - \alpha)$-quantile of the non-conformity scores is increased with higher estimation error.

### E.5    INTERPRETATION OF PARAMETER $M$

Our optimization requires the specification of a parameter $M$, denoting a bound on the propensity estimation error. One can view the parameter $M$ as a type of sensitivity parameter. Therefore, we follow former work in causal inference and propose to incorporate domain knowledge to specify the parameter $M$ (e.g., Frauen et al., 2024; Tan, 2006). Another way of making use of the parameter $M$ is to observe how the intervals change for varying $M$. This indicates how much effect the propensity misspecification has on the prediction interval and can help in making reliable decisions. A third option to calibrate $M$ is to employ measures for epistemic uncertainty on top of the propensity estimate when there is no domain knowledge for specifying $M$.

### E.6    A NOTE ON THE STABILITY OF OUR METHOD

In our experiments, one can observe some instability for certain privacy budget and intervention combination. This is likely due to the fact that the CP coverage guarantees are only *marginal*. Therefore, we might experience under- or over-coverage. However, we note that across all runs, our method, on average, achieves the desired coverage. These instabilities only occur in single settings. Furthermore, the variance in coverage of our method is much lower than of the coverage variance of the baselines. A more in-depth analysis of the stability of the proposed method is left for future work.

# F EXPERIMENTATION DETAILS

## F.1 SYNTHETIC DATASET GENERATION

We consider two different propensity and outcome functions. In each setting, we assign two types of interventions: a known propensity shift of $\Delta = 1, 5, 10$, i.e., three soft interventions $a^* = a + \Delta$, and the point interventions $a^* \in \{1x, 5x, 10, \}$ given the confounder $X = x$.

We generate synthetic datasets for each setting. Specifically, we draw each 2000 train, 1000 calibration, and each 1000 test samples per intervention from the following structural equations. Dataset 1 is given by

$$X \sim \text{Uniform}[1, 4] \quad \text{(integer)}$$
$$A \sim p \cdot \text{Uniform}[0, 5X) + (1 - p)\text{Uniform}[5X, 40], \quad p \sim \text{Bernoulli}(0.3)$$
$$Y \sim \sin\left(\frac{\pi}{6}(0.1A - 0.5X)\right) + \text{Normal}(0, 0.1),$$

and dataset 2 by

$$X \sim \text{Uniform}[1, 4] \quad \text{(integer)}$$
$$A \sim \text{Normal}(5X, 10)$$
$$Y \sim \sin\left(\frac{\pi}{2}(0.1A - 0.1X)\right) + \text{Normal}(0, 0.1),$$

## F.2 MEDICAL DATASET

We use the MIMIC-III dataset (Johnson et al., 2016), which includes electronic health records (EHRs) from patients admitted to intensive care units. From this dataset, we extract 8 confounders (heart rate, sodium, blood pressure, glucose, hematocrit, respiratory rate, age, gender) and a continuous treatment (mechanical ventilation) using an open-source preprocessing pipeline (Wang et al., 2020). From each patient trajectory in the EHRs, we sample random time points and average the value of each variable over the ten hours before the sampled time point. We define the variable blood pressure after treatment as the outcome, for which we additionally apply a transformation to be more dependent on the treatment and less on the blood pressure before treatment. We remove all patients (samples) with missing values and outliers from the dataset. Outliers are defined as samples with values smaller than the 0.1th percentile or larger than the 99.9th percentile of the corresponding variable. The final dataset contains 14719 samples, which we split into train (60%), val (10%), calibration (20%), and test (10%) sets.

## F.3 IMPLEMENTATION DETAILS

Our experiments are implemented in PyTorch Lightning. We provide our code in our GitHub repository.

We limited the experiments to standard multi-layer perception (MLP) regression models, consisting of three layers of width 16 with ReLu activation function and MC dropout at a rate of 0.1, optimized via Adam. We did not perform hyperparameter optimization, as our method aimed to provide an agnostic prediction interval applicable to any prediction model. All models were trained for 300 epochs with batch size 32.

Our algorithm requires solving (non-convex) optimization problems through mathematical optimization. We chose to employ two interior-point solvers in our experiments: For the experiments with soft interventions that pose convex optimization problems, we use the solver MOSEK. For the hard interventions, which included non-convex problems, we used the solver IPOPT. Both solvers were run with default parameters.

# G    FURTHER RESULTS

We present further results from our experiments in Section 5. Specifically, we state the prediction performance of the underlying models $\phi$, discuss the scalability of our approach, and show the prediction intervals per covariate for various significance levels $\alpha$ and soft interventions $\Delta$ of our synthetic experiments on dataset 1 and dataset 2.

**Performance:** We first report the performance of the underlying prediction models $\phi$ for the synthetic datasets across 50 runs in Table G. The prediction model on the real-world dataset achieved a mean squared error loss of 1.2373.

|  | Dataset 1 | Dataset 2 | MIMIC |
|---|---|---|---|
| $\phi$ | 0.0216 (0.0056) | 0.9029 (0.3908) | 0.0141 (0.0057) |
| Ens. | 0.0094 ($2.1169e^{-5}$) | 0.0130 (0.0003) | - |

Table 4: Mean and standard deviation of MSE loss of prediction models $\phi$ across 50 runs.

We further report the width of the prediction intervals in our synthetic experiments in Table G. The width is important to assess the usefulness of the resulting prediction intervals. As the performance of the ensemble method is not comparable with the coverage of MC-Dropout and our CP method, we only compare the latter two methods with regard to the interval width.

|  | Dataset 1 | | Dataset 2 | |
|---|---|---|---|---|
| Delta | Ours | MC-Dropout | Ours | MC-Dropout |
| 1 | 0.3647 (0.1284) | 0.1938 (0.1170) | 0.4051 (0.1036) | 0.2897 (0.1480) |
| 5 | 0.4024 (0.2285) | 0.1653 (0.1103) | 0.4610 (0.2479) | 0.3036 (0.1455) |
| 10 | 0.4301 (0.2610) | 0.1639 (0.1080) | 0.6711 (0.8520) | 0.3235 ( 0.1445) |

Table 5: Mean and standard deviation of the resulting prediction intervals.

**Scalability:** Calculating the prediction intervals requires an iterative search for an optimal value $S^*$. Therefore, the underlying optimization problem must be fitted multiple times throughout the algorithm, potentially posing scalability problems. In our empirical studies, however, we did not encounter scalability issues. Importantly, we found that the average runtime of our algorithm on a standard desktop CPU is only 16.43 seconds.

**Prediction intervals:** In Figures 8, 9, and 10, we present the prediction bands given by our method and MC dropout on dataset 1. In particular, for confounder $X = 1$, our method shows a large increase in the uncertainty in the potential outcomes of treatments affected by the intervention. MC dropout does not capture this uncertainty.

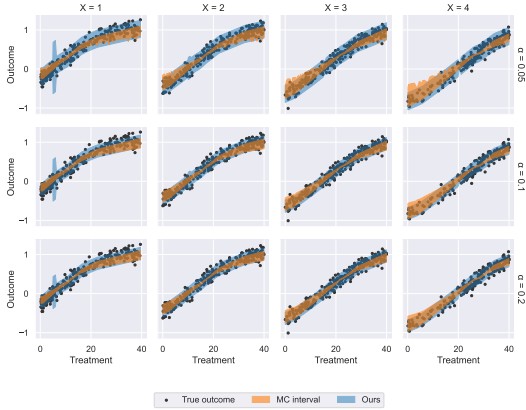

Figure 8: Prediction intervals for multiple significance levels $\alpha$ for the synthetic dataset 1 with intervention $\Delta = 1$.

In Figures 11 and 12, we present the prediction bands given by our method and MC dropout on dataset 2 for the soft interventions $\Delta = 1$ and $\Delta = 10$ (the results for $\Delta = 5$ were presented in the main paper). We observe that the prediction intervals for $\Delta = 10$ become extremely wide for high treatments. This aligns with our expectation, as data for high treatments in combination with low confounders is rare or even absent in the dataset. Thus, the expected uncertainty is very high.

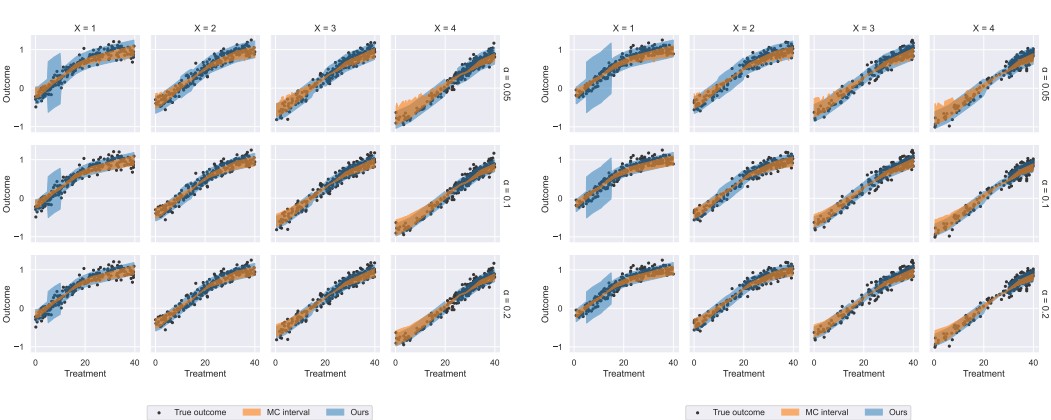

Figure 9: Prediction intervals for multiple significance levels $\alpha$ for the synthetic dataset 1 with intervention $\Delta = 5$

Figure 10: Prediction intervals for multiple significance levels $\alpha$ for the synthetic dataset 1 with intervention $\Delta = 10$.

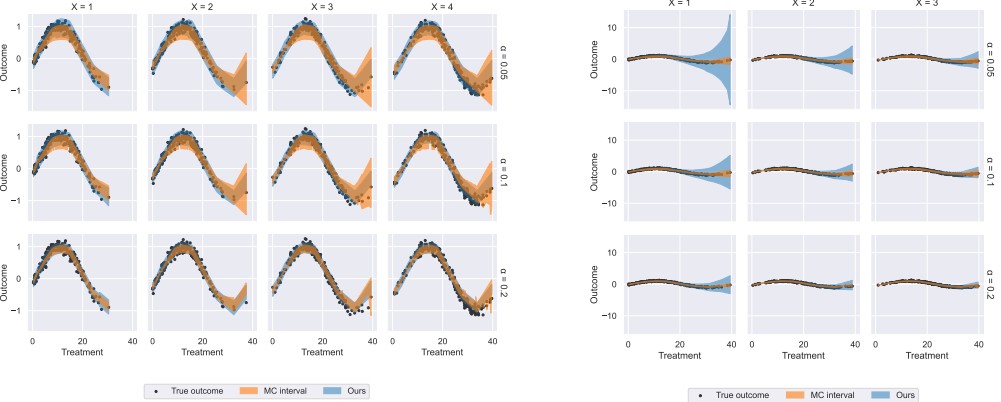

Figure 11: Prediction intervals for multiple significance levels $\alpha$ for the synthetic dataset 2 with intervention $\Delta = 1$

Figure 12: Prediction intervals for multiple significance levels $\alpha$ for the synthetic dataset 2 with intervention $\Delta = 10$.

