# OpenReview forum: "Conformal prediction for causal effects of continuous treatments"
_ICLR.cc/2025/Conference — Submitted to ICLR 2025_

### Official Review · Reviewer_nAyf · 2024-10-23

**Soundness:** 2
**Presentation:** 3
**Contribution:** 3
**Rating:** 5
**Confidence:** 3

**Summary:**

The paper applies the framework introduced by Gibbs et al. (2023) to produce prediction intervals, with a finite-sample coverage guarantee, for potential outcomes of continuous treatments. Gibbs et al. (2023) can provide certain types of conditional coverage guarantees by reformulating the coverage as coverage over a class of covariate shifts. Gibbs et al. (2023) already showed that given an appropriate choice of $\mathcal{F}$, that their prediction set achieves coverage under covariate shift, so long that the covariate shift can be represented as, $P_X$ titled by $f$, and $f \in \mathcal{F}$.

This work postulates the causal effect of continuous treatments as soft and hard interventions.  A soft intervention can be seen as a value shift in the generalized propensity score. A hard intervention can be seen as shifting the generalized propensity score to a Dirac delta function to the desired treatment value. In the paper, they define an appropriate function class for the different interventions and deal with their specific covariate shifts. For the hard intervention setting, they also ensure that the function class can deal with an estimated generalized propensity score with a bounded estimation error of the propensity function.

They provide some experiments on synthetic datasets (from their own) and medical ICU datasets, benchmarking their prediciton intervals against an MC dropout and Ensemble approach.

[1] I. Gibbs, J. J. Cherian, and E. J. Candès, “Conformal Prediction With Conditional Guarantees,” Dec. 20, 2023, arXiv: arXiv:2305.12616. doi: 10.48550/arXiv.2305.12616.

**Strengths:**

- **Clear Motivation:** The paper's objective is well-placed within causal machine learning, particularly in high-stakes areas like medicine, where uncertainty quantification is critical for decision-making.
- **Novel Contribution:** The authors propose a novel method for deriving conformal prediction intervals in the presence of continuous treatments and unknown propensity scores. This is a significant contribution, as most prior work focuses on binary or discrete treatments.

**Weaknesses:**

- **Terminology Issues:** Terms such as "finite-sample predictive intervals" should be revised to "finite-sample validity guarantees" to align with established terminology in the field. Additionally, (line 090) “strong finite-sample mathematical guarantees” should be “finite-sample validity guarantees”. Similarly, the frequent use of the term "exact" in relation to conditional coverage is misleading, as exact conditional coverage is known to be unattainable. I also assume you do not provide exact coverage in the context of exact and conditional coverage.
- **Relation to Lei and Candes (2021):** In Figure 2, and the appendix, the paper claims that the work of Lei and Candes can not deal with an unknown propensity, this is not true it defines and proves a coverage gap based on the error in the estimation of the likelihood ratio’s (= propensity in the case of treatment effect) in weighted CP.
- **Relation to Gibbs et al. (2023): T**he paper mentions the paper of Gibbs et al. (2023) in that it reformulates the split CP as an augmented quantile regression and the use of Lemma 1. But throughout the work, several notions from this work are taken, such as the tilting of the propensity function (lines 210-215), Theorem 1 (lines 262-276) and Theorem 2 (lines 352-371) which is heavily inspired by this work as it can even be seen in the borrowed notation of Section 4 of Gibbs et al. (2023), Algorithm 1 similarly with Algorithm 1 of Gibbs et al. (2023).
- **Under-coverage in Empirical Results**: The experimental results show some under-coverage, particularly for specific significance levels (e.g., α = 0.1). This variability raises concerns about the method's robustness, and the number of runs (10) in some experiments is too low to draw reliable conclusions. More runs would improve the statistical robustness of the findings.
- **Computational Complexity**: While the paper introduces an algorithm, the appendix discusses the method's computational efficiency. It points out that complexity can be high due to the optimization solver. This should be included in the limitations. Additionally, you should include your experiments' fitting and inference times in the paper, as this could indicate if computational complexity is an issue in practice.
- **Too simple simulation experiments: T**he proposed synthetic experiments are way too simplistic, with only one discrete confounder, continuous treatment, and continuous outcome. This is not in line with other treatment effect literature; we cannot adequately evaluate the method with these experiments.
- **Baseline**: The choice of baseline methods, particularly MC dropout, may not be ideal for comparison. Since these approaches only quantify epistemic uncertainty, which is probably minimal due to the simple experiment settings, consider a benchmark approach that quantifies both aleatoric and epistemic uncertainty, or at least aleatoric such as the Gaussian process or maximum likelihood estimator. The MLE can be easily combined with MC dropout to get both aleatoric and epistemic to get a reasonable benchmark. Also, you did not evaluate vanilla CP without considering covariate shift as a benchmark.

**Questions:**

- What is your added contribution to the work of Gibbs et al. (2023), which you leverage in this work?
- Can you explain the variability in empirical coverage across different significance levels, particularly for α = 0.1? Do you plan to increase the number of runs or experiments to solidify your claims?
- Could you indicate how long it takes to run these experiments?
- I think there is a typo on line 476 that should be Table 1, right?
- In your synthetic experiments, why are the hard interventions related to x and not just hard interventions?
- Related to Assumption 1 (line 295-299), does this assumption mean that the estimation error needs to be the same for each X given an intervention $a$, otherwise should it not be $c_a(x)$ instead of $c_a$.
- Related to Theorem 2, what is the meaning of resulting $\sigma$ and $c_a$ of the optimization problem.
- Related to Lemma 4 (lines 712-736):
    - I think you need to clarify that this is only relevant for absolute residual non-conformity score.
    - Next, I think the proof is complete. You do not prove the connection between the Bonferonni corrected significance level and the targeted significance level.
    - In my opinion, the proof for this lemma is even elementary since it is just an application of the Bonferonni correction.

---

> ### Author Response · Authors · 2024-11-24
> **Response to reviewer nAyf**
>
> Dear Reviewer nAyf,
>
> Thank you for your detailed feedback on our manuscript. We took all your comments at heart and improved our manuscript accordingly. We **updated our PDF** and highlighted all key changes in **blue color**.
>
> ### Answer to weaknesses:
>
> - **Terminology:**
> Thank you for helping us to improve the reader’s understanding of our manuscript by clarifying our employed terminology.
>
>     **Action:** We improved the terminology according to your suggestions in the updated version of our manuscript.
>
> -  **Relation to Lei and Candes (2021)**:
> Thank you for raising this concern. The reviewer is correct in that Lei and Candès (2021) provide an approach for CP with unknown propensity score through specifying a so-called coverage gap. Specifically, Lei & Candés (2021) consider the estimated propensity by incorporating the estimation error as a TV-distance term in the coverage guarantees. However, the TV-distance assumption contains several drawbacks compared to our assumption based on density ratios (DR): The bounded DR assumption is more universal. For example, the DR assumption implies boundedness of the TV-distance (see Bretagnolle and Huber inequality) or f-divergences.
> For large TV-distances (close to 1), Lei & Candes can only construct informative intervals with a very limited coverage $\alpha \in (0, 1 - TV)$), since, otherwise, the method yields non-informative intervals. In contrast, thanks to our DR assumption, **we can infer informative intervals with any coverage  ($\alpha \in (0, 1)$) for any $M$**.
>
>     We acknowledge that our statement is very strong and unclear, and we apologize for the confusion. We thus rewrote the corresponding passages to include the explanations above.
>
>     **Action:** We **updated the corresponding passages in the updated version of our manuscript**.
>
> - **Relation to Gibbs (2023)**:
> The reviewer is correct that our work is built upon previous work *outside the field of causal inference* from Gibbs (2023). We apologize if this did not become clear in our manuscript. However, we note that our contribution lies in combining two research areas: causal inference and conformal prediction. Furthermore, Gibbs (2023) cannot be directly applied to our setting but only serves as an underlying tool which has to carefully be adapted to our setting (i.e., the propensity estimation and smoothing).
>
>     **Action:** We referred to Gibbs (2023) more frequently in the appropriate parts of our manuscript. We also spelled out the differences more clearly.
>
> - **Under-coverage of empirical results:**
> Thank you for letting us explain the variability in the coverage and for spotting a typo regarding the number of runs we conducted in our experiments. The reviewer is correct in that Fig. 4 and 5 show some instability for certain cases. This is likely due to the fact that the CP coverage guarantees are only *marginal*. Therefore, we might experience under- or over-coverage. However, we note that across all runs, our method, on average, achieves the desired coverage. These instabilities only occur in single settings. Furthermore, the variance in coverage of our method is much lower than the coverage variance of the baselines.
>
>     Additionally, we would like to apologize for our mistake in stating the number of runs (10) in the text. This was due to inattention when transferring the text from a former version of the manuscript. Observe that in the figure descriptions, we **reported the correct number of runs (50)**.
>
>     **Action:** We **added a discussion on the stability of our method** to the updated version of the manuscript (see **new Supplement C**)
>
> - **Computational complexity:**
> Thank you for raising this concern. As stated, our optimization procedure can be computationally expensive. Therefore, the proposed method might not be applicable to quantifying uncertainty in very high-dimensional queries. We agree that information about the time needed for calculating the prediction intervals is important to judge its benefit to large real-world applications.
>
>     **Action:** We **stated the time for calculating the prediction intervals** in the updated version of our manuscript (see Supplement G). Furthermore, we added a statement on the computational complexity to the limitations section of our paper.

---

> ### Author Response · Authors · 2024-11-24
> **Response to reviewer nAyf (continued)**
>
> - **Experiments:**
> Thank you for giving us the chance to show the applicability of our method to higher dimensional datasets and show its superiority over further baselines. However, we note again that by choosing low-dimensional datasets, we can visually inspect the prediction intervals. Nevertheless, We later also show that our method scales to high-dimensional settings on actual medical datasets. To evaluate our method further, **we conducted additional experiments on the semi-synthetic TCGA**, which includes _high-dimensional gene expressions as confounders_. Furthermore, **we evaluate our method against additional baselines**. Specifically, we compare our method additionally against (i) vanilla CP and (ii) a Gaussian process estimator to show the usefulness and necessity of considering the propensity shift in CP and to assess the aleatoric uncertainty in our experiments.
>
> **Action:** We will report our new results in our **new Supplement C** of our updated manuscript.
>
>
> ### Answer to questions:
>
> 1., 2., 3. See our answers to the respective questions in “weaknesses”
>
> - **Typo:**
> Thank you for noting our typo.
>
>     **Action:** We corrected it in the updated version of our manuscript.
>
> - **Hard interventions in synthetic experiments:**
> The hard intervention in the synthetic experiments was designed to depend on the covariates as this is more realistic in practice. For example, a doctor would never consider giving the same medication dosage to a child as to an adult. Therefore, we are interested in prediction intervals for interventions which depend on the covariates.
>
>
> -  **Estimation error:**
> Thank you for raising this question. Assumption 1 does not imply that the estimation error has to be the same for all samples. We only assume an overall upper bound. Therefore, the parameter $c_a$ does not depend on the covariates $x$.
>
>
> - **Optimal parameters:**
> Thank you for giving us the chance to explain the meaning of the optimal parameters in more detail. The optimal parameter $\sigma^{\ast}$ represents a trade-off between the uncertainty in the prediction and the uncertainty in the interval construction: A small $\sigma^{\ast}$ resembles the propensity of the hard intervention best. Thus, with sufficient or even infinite data close to $a^{\ast}$, we could construct the narrowest CP interval. However, the smaller $\sigma$, the less data close to $a^{\ast}$ will be available to calculate the prediction interval in practice. As a result, many calibration data samples will be strongly perturbed during the calculation, which increases the uncertainty and, thus, the interval size. The parameter $c_a$ allows us to incorporate the estimation error in the propensity score. It represents a weighting of the propensity shift such that the $(1-\alpha)$-quantile of the non-conformity scores is increased with higher estimation error.
>
>     **Action:** We present a **discussion of the optimal parameters** in the updated version of our manuscript (see **Supplement D4**)
>
> - **Lemma 4:**
> Thank you for noting our precision. The reviewer is correct that Lemma 4 only applies if the non-conformity score equals the absolute residual. However, we note that this is the setting we considered throughout our paper. We apologize for being imprecise regarding the setting in Lemma 4.
>
>     **Action:** We **stated the setting in Lemma 4 more precisely** in the updated version of our paper.

---

> > ### Comment · Reviewer_nAyf · 2024-11-25
> >
> > Thank you for addressing the terminology issues, typos, and clarifications on parameter definitions. While these revisions improve the paper's presentation, I remain concerned about several key points:
> >
> > 1. **Omission of Lei and Candes from Figure 2:** Simply removing Lei and Candes work from Figure 2 is not an adequate way to address its relevance. Lei and Candes provide a seminal contribution to causal inference and conformal prediction, particularly their ability to handle unknown propensities with finite-sample exact guarantees. Their method is foundational, and excluding it from both Figure 2 and the main discussion undermines the context in which your approach operates. Including their work is crucial for a fair comparison, especially given its applicability to problems involving unknown propensity scores.
> >
> > 2. **Claims about informative intervals for any error in the estimation of the propensity:** While you acknowledge Lei and Candes reliance on TV-distance, your claim of producing informative intervals with any coverage for any $M$
> > remains unsupported. Without upper bounds, the claim is speculative. As you note yourself, higher estimation errors lead to increased $(1-\alpha)$-quantiles of conformity scores, which inherently reduces informativeness. This issue is particularly significant since it directly challenges the reliability of your method under high-error conditions. I would urge you to provide theoretical analysis or empirical evidence to back this claim or rephrase it to reflect the limitations
> >
> > 3. **Additional experiments:**  I did not find the additional experiments on the semi-synthetic dataset or the evaluation of Gaussian processes as promised in your response. Including these experiments is essential for validating your approach under different conditions and for providing a complete evaluation. The absence of these results weakens the empirical support for your claims.
> >
> > 4. **Practicality and computational complexity:** Your rebuttal does not fully address concerns regarding the practicality of the method, particularly in terms of computational complexity. While you suggest that the complexity resides in the solving operation, no details are provided about its scalability or real-world feasibility. This lack of transparency is concerning, as the solving operation could impose significant computational costs. I strongly recommend including an analysis of the time complexity and empirical evaluations demonstrating the scalability of your method.
> >
> > 5. **Clarification of differences with Gibbs et al. (2023):** Your response still does not clearly delineate how your approach differs from Gibbs et al. (2023). Is the distinction primarily limited to propensity estimation and smoothing of hard interventions? If so, this distinction should be explicitly discussed and justified in the manuscript. A deeper exploration of these differences would help clarify the novelty and unique contributions of your work.

---

> > > ### Author Response · Authors · 2024-11-27
> > >
> > > Dear reviewer nAyf,
> > >
> > > Thank you for taking the time to read our rebuttal and provide further constructive feedback. We apologize that some parts of our rebuttal did not sufficiently address your concerns or were not stated precisely enough. We are fully committed to remedying any remaining questions, and we have thus improved our work in the following ways. We marked our new revised parts in **red color** in the updated version of our manuscript.
> > >
> > > - **More extensive discussion of Lei and Candes from Figure 2:**
> > >
> > >     We apologize that we have discussed the work by Lei and Candes not as sufficiently as we should have done. We thus followed your feedback closely and discussed the work by Lei and Candes more extensively throughout our entire paper. In particular, **we updated Fig. 2 to include Lei and Candes**, so that we now also differentiate the methods based on treatment type. Furthermore, we have included a new discussion of their work in the related work section of our main paper (see our revised Section 2) and Supplement E.1. We find that your suggestion has greatly helped us to make the connection to existing works clearer and to spell out the novelty of our method.
> > >
> > >     **Action:** We provided a more extensive discussion of the method by Lei and Candes in our main paper (see our revised **Section 2** and our revised Fig. 2) and **Supplement E.1**.
> > >
> > > - **Informativeness of intervals:**
> > >
> > >     Thank you for noting the ambiguity of our usage of the word “informative”. Upon reading your comment, we realized that the term might be interpreted in different ways: (a) “informative” in the sense that the validity/faithfulness of the intervals (in terms of width) for a guaranteed confidence level of $(1-\alpha)$ is no longer ensured; and (b) “informative” in the sense that the width leads to guarantees that are not meaningful/useful in practice. We thus revised our paper to avoid potential ambiguity. In particular, we rewrote the respective parts in our manuscript and used more precise terminology (e.g., “width” as technical terminology).
> > >
> > >     **Action:** We rewrote the respective parts in our manuscript where we now avoid the term “informative” and instead use more precise terminology.
> > >
> > >
> > > - **Additional experiments:**
> > >
> > >     We apologize that we did not report the results in the first revision of our manuscript. We have now added them in **red color**.
> > >
> > >     **Action:** We **add vanilla CP and a GP-Regression as new baselines** to our experiments (see **Section 5**). Furthermore, we report the results on the semi-synthetic TCGA dataset in **Supplement C**. The results confirm the effectiveness of our method.
> > >
> > >
> > > - **Computational complexity:**
> > >
> > >     Thank you for your comment. We are unsure whether we have fully understood your question. So far, we have provided a **complexity analysis** of our algorithm (**Supplement B**). We have also provided a **discussion of the scalability** of our method. Therein, we also report the **average runtime** in our empirical evaluation (**Supplement F**). We hope that this addresses your question. Should you think that additional explanations or analyses are helpful, please let us know so – we are more than happy to provide them.
> > >
> > >
> > > - **Clarification of the differences to Gibbs et al. (2023):**
> > >
> > >     Our approach is primarily a contribution to the literature of causal inference, not to the pure conformal prediction. We thus provide a framework that is deeply rooted to solve the unique challenges of causal inference (e.g., propensity score estimation, hard/soft interventions). Of note, the challenges vary greatly depending on whether (a) binary treatments or (b) continuous treatments are considered. Our work is located in the latter setting. In contrast, the work by Lei & Candes is focused on a different setting, namely, the former one. Lei & Candes thus built upon weighted conformal prediction (Tibshiriani, 2019) to provide CP intervals for binary treatments. In contrast, we built upon the approach proposed by Gibbs (2023) to provide CP intervals for continuous treatments. Still, we offer a new, non-trivial framework to address the underlying challenges that come from computing CP intervals in a causal inference setting (e.g., propensity score estimation, hard/soft interventions), which we regard as our main novelty and which is of immediate practical relevance (e.g., in personalized medicine).
> > >
> > >     **Action:** We added a further discussion on the differences to Gibbs (2023) in **Supplement E.1**.

---

> > > > ### Comment · Reviewer_nAyf · 2024-11-27
> > > >
> > > > Thank you for the detailed response. Some of my concerns have been clarified, and I have revised my score to 5. However, I still have a few remaining issues regarding the experimental setup and reporting:
> > > >
> > > > - **Additional benchmark approaches:** The results presented for V-CP and GP appear cherry-picked. Could you provide results for all datasets and confidence levels to ensure a comprehensive evaluation? A complete table or appendix would help improve transparency.
> > > >
> > > > - **Semi-synthetic experiments (1):** The choice of the treatment effect in the TCGA dataset seems highly unconventional and overly simplistic. Could you explain the rationale behind this choice? Additionally, how is the response modeled? The current description of the semi-synthetic experiment setup feels incomplete.
> > > >
> > > > - **Semi-synthetic experiments (2):** This dataset provides an opportunity to evaluate your approach with a more realistic number of covariates (e.g., 20, 50, 100). Why not adopt setups similar to other works on continuous treatment effects, such as Bica et al. (2020), which also use the TCGA dataset? Aligning with these established setups could improve trust in the simulation experiments.
> > > >
> > > > - **Semi-synthetic experiments (3):** Why were the other baseline approaches not benchmarked in this setting?
> > > >
> > > > - **Time complexity:** The statement about time complexity, "the average runtime of our algorithm on a standard desktop CPU is only 16.43 seconds," lacks clarity. Does this refer to a single run, intervention, or example? For which experiment was this measured? Additionally, how does runtime scale with an increasing number of covariates or samples? A more detailed analysis of time complexity would be helpful.

---

> > > > > ### Author Response · Authors · 2024-12-02
> > > > > **Response to comment by Reviewer nAyf**
> > > > >
> > > > > Thank you for your additional feedback and for raising your evaluation score for our manuscript. We are happy to answer your remaining questions and will include all clarifications in the next update of our paper as well as in the camera-ready version of our paper.
> > > > >
> > > > > - **Additional benchmark approaches:** We agree with the reviewer that the results look unusual. However, they are not cherry-picked. Rather, the baselines are ineffective (compared to our method) for good reasons. As stated in the text, V-CP cannot achieve **any** valid prediction interval across the experiments. The GP is only able to capture the true potential outcome in the prediction intervals for small distribution shifts (∆ = 1) on dataset 2. In all other settings, GP can neither achieve valid prediction intervals.
> > > > >
> > > > >     For the sake of completeness, we present each one table reporting the mean empirical coverage of all baselines on both dataset 1 and 2 below. Every cell includes three entries that denote the mean empirical coverage for the three confidence levels: 0.05, 0.1, and 0.2.
> > > > >
> > > > > Dataset 1
> > > > > | Intervention | MC-Dropout | Ensemble | Vanilla-CP| Gaussian process| Ours |
> > > > > | :--------------- | :--------------: | :--------------: | :--------------: | :--------------: | :--------------: |
> > > > > |$\Delta = 1$ | 0.56 / 0.54 / 0.48 | 0.05 / 0.05 / 0.04 | 0.00 / 0.00 / 0.00 | 0.00 / 0.00 / 0.00 | 0.97 / 0.86 / 0.83  |
> > > > > |$\Delta = 5$ | 0.50 / 0.47 / 0.41 | 0.05 / 0.05 / 0.04 | 0.00 / 0.00 / 0.00 | 0.00 / 0.00 / 0.00 |  0.93 / 0.84 /  0.81 |
> > > > > |$\Delta = 10$ | 0.51 / 0.48 / 0.42 | 0.05 / 0.04 / 0.03 | 0.00 / 0.00 / 0.00 | 0.00 / 0.00 / 0.00 |  0.94 / 0.87 / 0.83 |
> > > > >
> > > > >
> > > > > Dataset 2
> > > > > | Intervention | MC-Dropout | Ensemble | Vanilla-CP| Gaussian process| Ours |
> > > > > | :--------------- | :--------------: | :--------------: | :--------------: | :--------------: | :--------------: |
> > > > > |$\Delta = 1$ | 0.67 / 0.65 / 0.59 | 0.10 / 0.08 / 0.06 | 0.00 / 0.00 / 0.00 | 0.13 / 0.13 / 0.08  | 0.97 / 0.93 / 0.81 |
> > > > > |$\Delta = 5$ | 0.69 / 0.67 / 0.60 | 0.10 / 0.09 / 0.07 | 0.00 / 0.00 / 0.00 | 0.00 / 0.00 / 0.00 |  0.97 / 0.93 / 0.81 |
> > > > > |$\Delta = 10$ | 0.67 / 0.65 / 0.59 | 0.12 / 0.10 / 0.07 | 0.00 / 0.00 / 0.00 | 0.00 / 0.00 / 0.00 | 0.94 / 0.88 / 0.78 |
> > > > >
> > > > >
> > > > > - **Semi-synthetic experiments:**
> > > > > We apologize that our description of the data-generating process was not sufficient. We are happy to provide further details.
> > > > >
> > > > >     In our semi-synthetic experiment, we model the outcome as a linear combination of the gene expressions. We add a treatment effect that is  linear in the treatment. We can view the treatment as the duration or intensity of radiotherapy, and its effect on the tumor size as the outcome. For transparency, all details for the semi-synthetic datasets are in our anonymized GitHub repository. We will present a detailed description of the data-generating process in the camera-ready version of our manuscript.
> > > > >
> > > > >    We further analyzed our method on the popular benchmarking version of the TCGA dataset presented in Bica et al. (2020). We focus only on one treatment, while Bica focuses on multiple treatments, so we proceeded by running our experiments on treatment setting 1 in Bica (2020). Again, we consider the three different confidence levels $\alpha = 0.05,0.1,0.2$ and two interventions $\Delta = 0.1, 0.3$.
> > > > >
> > > > > TCGA following treatment and outcome modeling presented in Bica (2020)
> > > > > | Intervention | MC-Dropout | Ensemble | Vanilla-CP| Gaussian process| Ours |
> > > > > | :--------------- | :--------------: | :--------------: | :--------------: | :--------------: | :--------------: |
> > > > > |$\Delta = 0.1$ | 0.02 / 0.03 / 0.02 | 0.14 / 0.13 / 0.10 | 0.00 / 0.00 / 0.00 | 0.00 / 0.00 / 0.00 | 0.96 / 0.88 / 0.77 |
> > > > > |$\Delta = 0.3$ | 0.16 / 0.01 / 0.01 | 0.08 / 0.06 / 0.06  | 0.00 / 0.00 / 0.00 | 0.00 / 0.00 / 0.00 | 0.95 / 0.89 / 0.76 |
> > > > >
> > > > >    We have also compared our method against the baselines. To this end, we have additionally computed all **four baselines** on the semi-synthetic dataset, which we present below. Every cell includes three entries that denote the mean empirical coverage for the three confidence levels: 0.05, 0.1, and 0.2. Again, we find that our proposed method is best.
> > > > >
> > > > > | Intervention | MC-Dropout | Ensemble | Vanilla-CP| Gaussian process| Ours |
> > > > > | :--------------- | :--------------: | :--------------: | :--------------: | :--------------: | :--------------: |
> > > > > |$\Delta = 0.5$ | 0.83 / 0.83 / 0.79 |  0.07 / 0.06  / 0.05  | 0.00 / 0.00 / 0.00 | 0.00 / 0.00 / 0.00 | 0.97 / 0.90 / 0.80|
> > > > > |$\Delta = 1.5$ | 0.87 / 0.85 / 0.80 |  0.09 / 0.07  / 0.05  | 0.00 / 0.00 / 0.00 | 0.00 / 0.00 / 0.00 | 0.97 / 0.95 / 0.77|
> > > > > |$\Delta = 1.5$ | 0.88 /  0.86 / 0.82 | 0.07 / 0.07  / 0.04  | 0.00 / 0.00 / 0.00 | 0.00 / 0.00 / 0.00 | 0.94 / 0.89 / 0.82|
> > > > >
> > > > >    **Action:** We will add the results from above and further explanation of the data- generating process to the camera-ready version of our manuscript.

---

> > > > > > ### Author Response · Authors · 2024-12-02
> > > > > > **Response to comment by Reviewer nAyf (continued)**
> > > > > >
> > > > > > - **Time complexity:**
> > > > > > Thank you for giving us the chance to recall and explain the time complexity in more detail.
> > > > > >
> > > > > >     **Average runtime:** The stated runtime relates to one run across the entire interventive dataset (the test dataset modified by the intervention) for one specific intervention averaged across all experiments (dataset, intervention, confidence level, and seed combinations). The experiments were run on standard office hardware in : 1x AMD Ryzen 7 PRO 6850U with 1x Radeon Graphics 2.70 GHz, 32 GB RAM. As you can see, our CP algorithms is thus seamlessly applicable to standard office hardware.
> > > > > >
> > > > > >     **Covariates**: The effect of an increasing number of covariates affects the computation time through the starting bounds $S_{up}$ and $S_{low}$ and thus through the prediction performance of the base model. If more covariates lead to better predictions, the complexity might even decrease in the covariates.
> > > > > >
> > > > > >     **Samples**: As stated in Supplement B, the computation complexity depends on the size of the _calibration dataset_ $n_c$. The calculation might become costly for large calibration datasets as the constrained optimization problem requires $n_c +1$ constraints. An increase in the number of training samples does not directly affect the computation time.
> > > > > >
> > > > > >     **Action:** We will add the discussion and details above to the camera-ready version of our manuscript.

---

### Official Review · Reviewer_wfcX · 2024-10-30

**Soundness:** 2
**Presentation:** 2
**Contribution:** 3
**Rating:** 5
**Confidence:** 4

**Summary:**

Conformal prediction for causal estimands is useful because it provides finite-sample coverage guarantees. Predicting potential outcomes is challenging due to the potentially unknown distribution shift between the training set and the test set, which is subject to intervention. This work extends conformal prediction under domain shift to address the challenges that are introduced when the treatment variable is continuous, and the causal estimand is a dose response.

**Strengths:**

The problem of conformal prediction of dose responses is clearly significant. The authors proposed an original formulation that uses Gaussian smoothing over the treatments and finds an optimal bandwidth for each problem instance. It is also important to consider soft interventions (by the authors' definition) in addition to hard interventions, since it is common in medicine and other settings to study incremental changes.

**Weaknesses:**

* I do not understand some core elements of the motivation. What is the intention behind the separation of challenge (a) and challenge (b)? Estimating the propensity score is more difficult with continuous treatments, but it is still a challenge with discrete treatments. By my understanding, the main challenge for bringing conformal prediction to continuous treatments is that you rarely see data points with the same treatment value, so you have to do some sort of smoothing. The motivating challenges do not mention this (unless I am misreading one of them).
 * I also found the roles of soft and hard interventions confusing. Under "Scenarios" in Section 4, the authors explain that soft interventions are relevant when the propensity is known, while hard interventions are reserved for when the propensity is unknown. This storyline is convenient for the mathematical derivations. However, I am not convinced: even if the propensity function is unknown, it can still be of interest to estimate the effect of increasing the treatment by a small amount on top of the existing policy. These cases have been studied before:
   * Many papers by Chernozhukov et al. that mention "average causal derivatives"
   * Marmarelis et al. "Policy Learning for Localized Interventions from Observational Data" in AISTATS 2024.
 * When the propensity / policy is unknown, the validity of the conformal prediction relies on a constant $M$ that bounds the error of the estimated propensity. This resembles a sensitivity analysis. It appears difficult to know which $M$ to use.
 * The results from the medical dataset do not seem convincing. It is good to see that intervals are wider in treatment regions that are less common in the data. However, the fact that intervals are wider than MC-dropout is not necessarily a good thing.

Minor notes.
 * "Quantile regression might yield non-unique solutions, so we later restrict the analysis to solvers invariant to the data ordering". Please expand on this sentence. How does ordering invariance help?
 * The algorithm text is small and quite dense.
 * An interpretation of the optimized $\sigma$ is provided in the supplement, but further discussion would be helpful. Please clarify its role in the optimization problem.

**Questions:**

* Consider building semi-synthetic experiments for stronger results.
 * How can $M$ of Assumption 1 be calibrated in practice, for tightness?
 * Do you assume some kind of continuity in the dose response?

---

> ### Author Response · Authors · 2024-11-24
> **Response to reviewer wfcX**
>
> Dear Reviewer wfcX,
>
> Thank you for your detailed feedback on our manuscript. We took all your comments at heart and improved our manuscript accordingly. We **updated our PDF** and highlighted all key changes in **blue color**.
>
> ### Answer to weaknesses:
>
> - **Motivation**:
> Thank you for giving us the chance to clarify the motivation for considering the different challenges in our manuscript. It is true that the problem of the unknown propensity score is present for discrete treatments as well as continuous treatments. These challenges are _not_ unique to the setting of continuous treatments. However, challenge (b) is often neglected in existing works for CP for discrete treatments. More importantly, we want to emphasize that our paper bridges two different fields. It especially targets readers from the field of treatment effect estimation who might not be familiar with CP. Therefore, we aim to highlight the challenges in CP for this setting for these readers.
>
>     **Action:** We have improved our motivation and now mention that some challenges also apply to discrete treatment settings but we highlight that unique to the continuous treatment setting is that we rarely see data points and thus must involve smoothing.
>
> - **Hard and soft interventions**:
> Thank you! Indeed, the missing setting could also be of interest by studying the effect of soft interventions, even if the propensity score is unknown. Upon reading your comment, realized that we were missing this case and should extend our paper accordingly. We thus extended our CP algorithm to this scenario. Specifically, we derive corresponding coverage guarantees for soft interventions on the estimated propensity score.
>
>     **Action:** We have clarified the applicability of soft interventions for both scenarios. Furthermore, we added **new Theorem 4** which provides coverage guarantees for soft interventions on the estimated propensity score.
>
>
> - **Constant M**:
> Thank you for raising this question. Indeed, one can view the parameter $M$ as a type of sensitivity parameter. This implies that one needs to make an assumption about the maximum error-bound. In employing this bound, we follow former work in causal inference which incorporates domain knowledge to specify the parameter $M$ (e.g., [1],[2],[3]). Another way of making use of the parameter $M$ is to observe how the intervals change for varying $M$. This indicates how much effect the propensity missspecification has on the prediction interval and can help in making reliable decisions. A third option to calibrate $M$ is to employ measures for epistemic uncertainty on top of the propensity estimate when there is no domain knowledge for specifying $M$.
>
>     **Action:** We added a discussion on $M$ in the updated version of our manuscript (see **new Supplement D.5**)
>
> - **Results on the medical dataset:**
> Thank you for giving us the chance to elaborate on the results of our experiment. We agree that only comparing the method based on width is difficult. To provide a better evaluation, we will further present results on semi-synthetic data (based on your suggestion) in the updated version of our manuscript. This allows us to also numerically evaluate the coverage of our method.
>
>
> ### Answer to minor notes:
>
> - **Non-unique solutions of quantile regression:**
> Thank you for noting that this might not be clear for the reader. Quantile regressions might have multiple optimal values, which can depend on the indices of the scores (e.g., see [8]). Therefore, we must restrict our theoretical analysis to solvers that are invariant to the data ordering. We note than commonly used solvers, such as interior point solvers, are invariant to the data ordering.
>
>     **Action:** We included more explanation to motivate the choice of our solver in the updated version of our manuscript.
>
> - **Size of algorithm text:**
> We thank the reviewer for noting that the text in our pseudo-code is difficult to read.
>
>     **Action:** We increased the size of the text in the updated version of our manuscript.
>
> - **Parameter $\sigma$:**
> Thank you for giving us the chance to elaborate further on the parameter $\sigma$. In the optimization problem, $\sigma$ acts as an optimization variable. Specifically, $sigma$ functions as a parameter describing the optimal trade-off between the uncertainty in the prediction and the uncertainty in the interval construction. By optimizing over $\sigma$, we make sure that we achieve the best approximation of the non-conformity scores and thus of the quantile of interest for constructing the prediction interval.
>
>    **Action:** We extended our discussion on $\sigma$ in the updated version of our manuscript

---

> ### Author Response · Authors · 2024-11-24
> **Response to reviewer wfcX (continued)**
>
> ### Answer to questions:
>
> - **Semi-synthetic experiments:**
> Thank you for noting that an evaluation on semi-synthetic data can be valuable to assess the performance of our method. We will thus **provide further results for the TCGA covariate dataset**.
>
>     **Action:** We  will **add semi-synthetic experiments** to the updated version of our manuscript.
>
> - **Calibration of M**:
> Please see our answer to your question on $M$ above.
>
> - **Smoothness of the dose-response curve**:
> Thank you for raising this questions. Indeed, we assume the dose-response curve to fulfill some smoothness criterion. This is common in causal inference with continuous treatments (e.g., [4],[5]). This also motivates the use of kernel smoothing for estimating the treatment effect. Again, this is common in causal treatment effect estimation (e.g., [6],[7]).
>
>     **Action:** We provide a discussion on the smoothness of the dose-response curve in **Supplement D.3**.
>
>
>
> [1]	Dennis Frauen, Fergus Imrie, Alicia Curth, Valentyn Melnychuk, Stefan Feuerriegel, and Mihaela van der Schaar. A neural framework for generalized causal sensitivity analysis. In International Conference on Learning Representations (ICLR), 2024.
>
> [2]	Nathan Kallus, Xiaojie Mao, and Angela Zhou. Interval estimation of individual-level causal effects under unobserved confounding. In International Conference on Artificial Intelligence and Statistics (AISTATS), 2019.
>
> [3]	Zhiqiang Tan. A distributional approach for causal inference using propensity scores. Journal of the American Statistical Association, 101(476):1619–1637, 2006.
>
> [4]	Patrick Schwab, Lorenz Linhardt, Stefan Bauer, Joachim M. Buhmann, and Walter Karlen. Learning counterfactual representations for estimating individual dose-response curves. In Conference on Artificial Intelligence (AAAI), 2020.
>
> [5]	Jonas Schweisthal, Dennis Frauen, Valentyn Melnychuk, and Stefan Feuerriegel. Reliable off-policy learning for dosage combinations. In Conference on Neural Information Processing Systems (NeurIPS), 2023.
>
> [6]	Edward H. Kennedy. Nonparametric causal effects based on incremental propensity score interventions. Journal of the American Statistical Association, 114(526):645–656, 2019.
>
> [7]	Lokesh Nagalapatti, Akshay Iyer, Abir, and Sunita Sarawagi. Continuous treatment effect estimation using gradient interpolation and kernel smoothing. In Conference on Artificial Intelligence (AAAI), 2024.
>
> [8]	Isaac Gibbs, John J. Cherian, and Emmanuel J. Candès. Conformal prediction with conditional guarantees. arXiv preprint, 2023

---

> > ### Comment · Reviewer_wfcX · 2024-11-26
> >
> > Thank you for your detailed rebuttal. I appreciate that some of my concerns were addressed sufficiently. However, in my view, the main issues with this manuscript still remain.
> >
> > For instance, it would be important to see semi-synthetic benchmarks in the main text. The authors promised to generate these results, but they have not appeared yet, so it is difficult to predict what they will look like. The results on the medical dataset have not changed, and so has my assessment of them.
> >
> > Overall, I also agree with Reviewer nAyf's main critiques.
> >
> > The clarifications to the manuscript were helpful and I look forward to a solid future revision.

---

> ### Author Response · Authors · 2024-11-29
>
> Dear reviewer wfcX,
>
> Thank you for taking the time to read our rebuttal and provide further constructive feedback. We are fully committed to remedying any remaining questions.
>
>
> In our manuscript, we have reported **experimental results** for our method on the **medical semi-synthetic** TCGA dataset (see **red color** in **Supplement C**).
> We now have also compared our method against the baselines. To this end, we have now additionally computed all **four baselines** on the semi-synthetic dataset, which we present below. Every cell includes three entries that denote the mean empirical coverage for the three confidence levels: 0.05, 0.1, and 0.2. Again, we find that our proposed method is best.
>
> | Intervention | MC-Dropout | Ensemble | Vanilla-CP| GP | Ours |
> | :--------------- | :--------------: | :--------------: | :--------------: | :--------------: | :--------------: |
> |$\Delta = 0.5$ | 0.83 / 0.83 / 0.79 |  0.07 / 0.06  / 0.05  | 0 / 0 / 0 | 0  / 0  / 0 | 0.97 / 0.90 / 0.80|
> |$\Delta = 1.0$ | 0.87 / 0.85 / 0.80 |  0.09 / 0.07  / 0.05  | 0 / 0 / 0 | 0 / 0  / 0 | 0.97 / 0.95 / 0.77|
> |$\Delta = 1.5$ | 0.88 /  0.86 / 0.82 | 0.07 / 0.07  / 0.04  | 0 / 0 / 0 | 0  / 0  / 0 | 0.94 / 0.89 / 0.82|
>
>
> **Action:** We will add the results from above to the camera-ready version of our manuscript.

---

### Official Review · Reviewer_xiCH · 2024-11-03

**Soundness:** 3
**Presentation:** 3
**Contribution:** 3
**Rating:** 6
**Confidence:** 4

**Summary:**

In the paper, the authors propose a Conformal-prediction (CP) based method to obtain the interval estimation of the potential outcomes under continuous treatments.
To achieve this, the authors follows (Gibbs et al., 2023; Romano et al., 2019) and reformulated split CP as an augmented quantile regression.
The authors derive finite-sample prediction intervals for cases where the propensity score function is known and unknown.
Moreover, they provide finite-sample theoretical guarantees.

**Strengths:**

1. They study a very important question in the causal inference and provide novel methods.
2. They provide the finite-sample theorem for their methods.
3. They clearly and accessibly present the research question and methods, using figures and concrete examples.

**Weaknesses:**

The authors do not have a good evaluation of their method in terms of numerical studies.

Firstly, the authors only focus on the empirical coverage, and claim that it should be the higher the better.
In my opinion, it seems not very reasonable.
If we simply choose the [-inf, inf] as the interval, it is always has the 100% empirical coverage.
A more balanced evaluation should consider both the interval length and empirical coverage. Additionally, the empirical coverage should closely match the nominal coverage for optimal results.


When checking the numerical results, I do believe the method in the implementation is not very stable under some cases.
For example, in Fig 4 ($\alpha=0.2$, 5), with only 50 repetitions, I can see many outliers in the boxplot, indicating there may be some issues there. Similar issue can be found in Fig 5 ($\alpha=0.1$, 1) and  ($\alpha=0.2$, 5).
Anyway, I know such unstableness may be inevitable for a complicated method, I think the authors should have some discussions on this issue.

**Questions:**

1. Major question is in **Weakness** part
2. I think the real data in the paper can not justify the superiority of authors' method well as I can see at range [0.6, 1], it yields not very reasonable prediction (blood pressure can be less than 0).
Also, the interval estimate is too wide to be meaningful.

---

> ### Author Response · Authors · 2024-11-24
> **Response to reviewer xiCH**
>
> Dear Reviewer xiCH,
>
>
> Thank you for your positive evaluation of our manuscript and your detailed feedback. We took all your comments at heart and improved our manuscript accordingly. We **updated our PDF** and highlighted all key changes in **blue color**.
>
> ### Answer to weaknesses
>
> - **Numerical evaluation**:
> Thank you. We followed your suggestion closely and improved our empirical evaluation as follows. In our updated manuscript, **we will add new experimental results on higher-dimensional covariates**, i.e., the TCGA dataset including high-dimensional gene expressions. Furthermore, **we evaluate our method against additional baselines**: (i) vanilla CP, without adjusting for the propensity shift, and (ii) a Gaussian process prediction.
>
>     **Action:** We will **add our new experimental results** to **new Supplement F** of our updated manuscript.
>
>
> - **New evaluation metric:**
> Thank you for this question. The interval [-inf,inf] would indeed be a very trivial but unuseful solution. To properly evaluate our method, **we now also report the width of the resulting prediction intervals**. However, we note that coverage is a more important metric than interval width. In practice, one can easily decide on the usefulness of the resulting interval by looking at the interval range, whereas one can never make any conclusion on the coverage of a returned interval which does not fulfill mathematical guarantees.
>
>     **Action:** We further evaluated our method in terms of the width of the prediction interval (see our new results in **Supplement G**).
>
>
> - **Stability of our method**:
> Thank you for pointing this out. The reviewer is correct in that Fig. 4 and 5 show some instability for certain cases. This is likely due to the fact that the CP coverage guarantees are only *marginal*. Therefore, we might experience under- or over-coverage. However, we note that, on average across all runs, _our method achieves the desired coverage_. These instabilities only occur in singular settings. Furthermore, the _variance in the coverage of our method is much lower than of the coverage variance of the baselines_.
>
>     **Action:** We added a discussion on the stability of our method to the updated version of the manuscript (see **new Supplement D.6**)
>
>
>
> ### Answer to questions
>
> - **New experiments on medical data**:
> Thank you. Upon reading your question, we realized that we have forgotten to rescale the outcome after the prediction. This led to very unreasonable results in terms of blood pressure. Furthermore, we realized that, from a medical perspective, estimating the effect of ventilation on blood pressure might not be of interest. Therefore, **we will present an updated experiment** in which we estimate the hematocrit level after ventilation instead of the blood pressure. We hope our new analysis is more meaningful and thus of larger interest.
>
>     **Action:** We will revise our experiment to provide an example that is more clinically relevant.

---

> ### Comment · Reviewer_xiCH · 2024-11-25
>
> Thanks for the authors' replies.
> I am overall satisfied with the rebuttal.

---

### Official Review · Reviewer_qNm2 · 2024-11-04

**Soundness:** 3
**Presentation:** 3
**Contribution:** 3
**Rating:** 6
**Confidence:** 2

**Summary:**

This paper proposes a conformal prediction method for continuous treatments, addressing the challenge of unknown propensity scores. It derives finite-sample prediction intervals, provides an algorithm for calculating them, and demonstrates effectiveness on synthetic and real datasets. This is the first method for continuous treatments with unknown propensity scores.

**Strengths:**

Clear writing & solid theoretical results

**Weaknesses:**

The reviewer is not an expert in conformal prediction for causal inference but would assume there are working targeting the studies setting: uncertainty quantification of causal effects for continuous treatment.

Equation 14 seems to use the kernel function to approximate the indicator function. Does this imply some smoothness assumptions?

The optimal parameter $\sigma^*$ represents a trade-off between the uncertainty in the prediction and the uncertainty in the interval construction. It is similar to the bandwidth approximation of the indicator function using a kernel function. Are there any empirical values ​​here, such as \sigma = cn^{-2/5} to  trade-off between the prediction and interval.

The method is demonstrated on specific datasets and scenarios. While the authors claim that their method scales to high-dimensional settings, further validation on a broader range of datasets and treatment types would strengthen the generalizability of the findings.

**Questions:**

See weakness.

---

> ### Author Response · Authors · 2024-11-24
> **Response to reviewer qNm2**
>
> Dear Reviewer qNm2,
>
>
> Thank you for your detailed feedback on our manuscript. We took all your comments at heart and improved our manuscript accordingly. We **updated our PDF** and highlighted all key changes in **blue color**.
>
>
> - **Existing work for uncertainty quantification for effects of continuous treatments**:
> Thank you for the question. Yes, there exist a variety of work for uncertainty quantification (UQ) in causal effect estimation for continuous treatments. Existing methods for UQ of causal quantities are often based on Bayesian methods (e.g., [1],[2],[3]). However, Bayesian methods **require the specification of a prior distribution** based on domain knowledge and are thus neither robust to model misspecification nor generalizable to model-agnostic machine learning models. Other methods **only provide asymptotic guarantees** (e.g. [4],[5]). The strength of conformal prediction, however, is to provide finite-sample uncertainty guarantees. Overall, **none of the methods** tackles the problem of distribution-free uncertainty quantification for causal effect estimation of continuous treatments in finite sample settings.
>
>     **Action:** We extended our related work to cite additional works on UQ in causal inferences but where we highlight the advantages of our conformal prediction (see our **revised Supplement D.1**).
>
>
> - **Smoothness assumption**: Thank you for raising this question. Indeed, Eq. 14 performs kernel smoothing to approximate the indicator function. As inherent to kernel smoothing, this requires sufficient smoothness of the underlying functions, which, in our case, is the propensity function. However, we note that this assumption is standard in causal effect estimation of continuous treatments (e.g. [6],[7]). Hence, when estimating treatment effects, interpolation and kernel smoothing of the outcome function are commonly employed (e.g., [8],[9]).
>
>     **Action:**: We highlight that the smoothness assumption is standard in causal inference for continuous treatments (see our **revised Section 3**). We further provide a discussion of the employed kernel-smoothing and the underlying smoothness assumption in **Supplement D.3**.
>
>
> - **Trade-off through the parameter $\sigma$**:
> Thank you for giving us the opportunity to provide more information on the trade-off imposed by the parameter $\sigma$. The optimal parameter represents a trade-off between the uncertainty in the prediction and the uncertainty in the interval construction: A small $\sigma$ resembles the propensity of the hard intervention best. Thus, with sufficient or even infinite data close to $a^{\ast}$ , we could construct the narrowest CP interval. However, the smaller $\sigma$, the less data close to $a^{\ast}$ will be available to calculate the prediction interval in practice. As a result, many calibration data samples will be strongly perturbed during the calculation, which increases the uncertainty and, thus, the interval size.
>
>     **Action:**: We added a discussion on the trade-off through parameter $\sigma$ in **Supplement D.4**.
>
>
> - **Empirical evaluation:**
> Thank you. We followed your suggestion closely and improved our empirical evaluation as follows. In our updated manuscript, **we added new experimental results on higher-dimensional covariates**, i.e., the TCGA dataset including high-dimensional gene expressions. Furthermore, **we evaluate our method against additional baselines**: (i) vanilla CP, without adjusting for the propensity shift, and (ii) a Gaussian process prediction.
>
>     **Action:** We **will add our new experimental results** to **new Supplement F** of our updated manuscript.

---

> > ### Author Response · Authors · 2024-11-24
> > **Response (continued)**
> >
> > [1]	Ahmed Alaa and Mihaela van der Schaar. Bayesian inference of individualized treatment effects using multi-task Gaussian processes. In Conference on Neural Information Processing Systems (NeurIPS), 2017
> >
> > [2]	Konstantin Hess, Valentyn Melnychuk, Dennis Frauen, and Stefan Feuerriegel. Bayesian neural controlled differential equations for treatment effect estimation. In International Conference on Learning Representations (ICLR), 2024.
> >
> > [3]	Andrew Jesson, Sören Mindermann, Uri Shalit, and Yarin Gal. Identifying causal-effect inference failure with uncertainty-aware models. In Conference on Neural Information Processing Systems (NeurIPS), 2020.
> >
> > [4]	Ying Jin, Zhimei Ren, and Emmanuel J. Candès. Sensitivity analysis of individual treatment effects:
> > A robust conformal inference approach. Proceedings of the National Academy of Sciences of the United States of America, 120(6), 2023.
> >
> > [5]	Jef Jonkers, Jarne Verhaeghe, Glenn van Wallendael, Luc Duchateau, and Sofie van Hoecke. Conformal Monte Carlo meta-learners for predictive inference of individual treatment effects. arXiv preprint, 2024.
> >
> > [6]	Patrick Schwab, Lorenz Linhardt, Stefan Bauer, Joachim M. Buhmann, and Walter Karlen. Learning counterfactual representations for estimating individual dose-response curves. In Conference on Artificial Intelligence (AAAI), 2020.
> >
> > [7]	Jonas Schweisthal, Dennis Frauen, Valentyn Melnychuk, and Stefan Feuerriegel. Reliable off-policy learning for dosage combinations. In Conference on Neural Information Processing Systems (NeurIPS), 2023.
> >
> > [8]	Edward H. Kennedy. Nonparametric causal effects based on incremental propensity score interventions. arXiv preprint, 2018.
> >
> > [9]	Lokesh Nagalapatti, Akshay Iyer, Abir, and Sunita Sarawagi. Continuous treatment effect estimation using gradient interpolation and kernel smoothing. In Conference on Artificial Intelligence (AAAI), 2024.

---

> > > ### Author Response · Authors · 2024-11-29
> > > **Follow-up on rebuttal**
> > >
> > > Dear reviewer qNm2,
> > >
> > > We hope our rebuttal sufficiently addressed all your concerns. If you have any more questions or concerns, we are happy to answer them as soon as possible. Please let us know if this is the case. Otherwise, we would highly appreciate it if you could raise your evaluation score for our manuscript.
> > >
> > > Best regards,
> > >
> > > The authors

---

### Author Response · Authors · 2024-12-02
**Comment to all reviewers and AC**

Dear reviewers, dear AC,

Thank you again very much for the constructive evaluation of our paper and your helpful comments! We addressed all of them in our rebuttal and the following discussion with the reviewers. However, we noticed that some reviewers have not yet responded to our latest updates and responses. If you have any remaining concerns, please do not hesitate to contact us. Otherwise, if you feel our responses have addressed your concerns satisfactorily, we would kindly appreciate it if you would consider updating the rating.
In summary, our main improvements are the following:

- We added **new empirical results** on a semi-synthetic medical dataset to show the applicability of our method to high-dimensional settings (see our **new Supplement C**).

- We compare our method against **additional baselines** (vanilla CP and a Gaussian process prediction) to assess the importance of accounting for the propensity shift (see our **Section 5** and tables direct answers to the reviewers). We find that our method is clearly effective.

- We provide **additional discussions** on the optimization parameters and the estimation error bound (see our **new Supplement E**). Furthermore, we improved the motivation for considering the different settings (hard and soft interventions; known and unknown propensity score) and elaborated on the differences to existing work for CP for binary or discrete treatments (see **Supplement E**).

- We added a **new Theorem**  for the special case when constructing prediction intervals for soft interventions under unknown propensity scores (see our **new Supplement A.3**).

As a result, we addressed all questions and comments from the review team. We incorporated all changes into the **updated version of our paper** or will upload all remaining changes to the camera-ready version. We highlighted all key changes in **blue and red color in our revised PDF**. Given these improvements, we are confident that our paper will be a valuable contribution to the literature on uncertainty quantification in causal effect estimation and a good fit for ICLR 2025.

---

### Meta-Review · Area_Chair_bzfS · 2024-12-21

**Metareview:**

This paper proposed a conformal prediction method for continuous treatments, addressing the challenge of unknown propensity scores. The problem is well motivated and the extension of conformal inference to continuous treatments is important. However, some main concerns remain:

 - The novelty and contribution compared with Gibbs et al. (2023) is limited, and theoretical results largely follows this paper.

 - There are concerns raised about the method's robustness and the empirical performance. For example, the authors report zero coverage for vanilla CP and GP methods, which suggests the presence of purely epistemic uncertainty and extremely heteroscedastic uncertainty and seems inconsistent with the data-generating processes. Additionally, there is a critical issue with the reported average width of prediction intervals for V-CP (claimed to be 0.0003). This value is mathematically improbable and is several orders of magnitude smaller than these theoretical lower bounds, which raises serious questions about the experimental methodology or reporting.

 - About utilities. The time complexity of the proposed approach is not very practical. The reviewers also suggested to have semi-synthetic benchmarks in the main text.

Given the above reasons, after discussion with the reviewers, we agreed it is not quite ready for publication and we'd encourage the authors to take into consideration all the feedback provided by the reviewers to strengthen their manuscript for resubmission.

**Additional Comments On Reviewer Discussion:**

Though part of reviewers' concerns have been nicely resolved during rebuttal, the main concerns from reviewers remain including the novelty and contribution compared with Gibbs et al. (2023), concerns about the method's robustness and the empirical performance, and practical utilities of the proposed method. After discussion with the reviewers, we agreed it is not quite ready for publication.

---

### Decision · Program_Chairs · 2025-01-22

Reject